



# Summary and synthesis of Changing Cold Regions Network (CCRN) research in the interior of western Canada – Part 2: Future change in cryosphere, vegetation, and hydrology

Chris M. DeBeer[1, 2], Howard S. Wheater[1, 2, 3], John W. Pomeroy[1, 2], Alan G. Barr[2, 4], Jennifer L. Baltzer[5], Jill F. Johnstone[6, 7], Merritt R. Turetsky[8, 9], Ronald E. Stewart[10], Masaki Hayashi[11], Garth van der Kamp[2], Shawn Marshall[12, 13], Elizabeth Campbell[14], Philip Marsh[15], Sean K. Carey[16], William L. Quinton[15], Yanping Li[2], Saman Razavi[2], Aaron Berg[17], Jeffrey J. McDonnell[2, 18], Christopher Spence[19], Warren D. Helgason[20], Andrew M. Ireson[2], T. Andrew Black[21], Bruce Davison[19], Allan Howard[22], Julie M. Thériault[23], Kevin Shook[1], and Alain Pietroniro[19]

[1]Centre for Hydrology, University of Saskatchewan, Saskatoon, Saskatchewan, Canada
[2]Global Institute for Water Security, University of Saskatchewan, Saskatoon, Saskatchewan, Canada
[3]Department of Civil and Environmental Engineering, Imperial College London, London, United Kingdom
[4]Climate Research Division, Environment and Climate Change Canada, Saskatoon, Saskatchewan, Canada
[5]Biology Department, Wilfrid Laurier University, Waterloo, Ontario, Canada
[6]Department of Biology, University of Saskatchewan, Saskatoon, Saskatchewan, Canada
[7]Institute of Arctic Biology, University of Alaska Fairbanks, Fairbanks, Alaska, United States
[8]Department of Integrative Biology, University of Guelph, Guelph, Ontario, Canada
[9]Department of Ecology and Evolutionary Biology, Institute of Arctic and Alpine Research, University of Colorado Boulder, Boulder, Colorado, United States
[10]Department of Environment and Geography, University of Manitoba, Winnipeg, Manitoba, Canada
[11]Department of Geoscience, University of Calgary, Calgary, Alberta, Canada
[12]Department of Geography, University of Calgary, Calgary, Alberta, Canada
[13]Environment and Climate Change Canada, Gatineau, Quebec, Canada
[14]Natural Resources Canada, Canadian Forest Service, Pacific Forestry Centre, Victoria, British Columbia, Canada
[15]Cold Regions Research Centre, Wilfrid Laurier University, Waterloo, Ontario, Canada
[16]School of Geography and Earth Sciences, McMaster University, Hamilton, Ontario, Canada
[17]Department of Geography, Environment and Geomatics, University of Guelph, Guelph, Ontario, Canada
[18]School of Geography, Earth & Environmental Sciences, University of Birmingham, Edgbaston, Birmingham, United Kingdom
[19]National Hydrology Research Centre, Environment and Climate Change Canada, Saskatoon, Saskatchewan, Canada
[20]Chemical and Biological Engineering, University of Saskatchewan, Saskatoon, Saskatchewan, Canada
[21]Faculty of Land and Food Systems, University of British Columbia, Vancouver, British Columbia, Canada
[22]Agriculture and Agri-Food Canada, Regina, Saskatchewan, Canada
[23]Centre ESCER, Department of Earth and Atmospheric Sciences, Université du Québec à Montréal, Montréal, Quebec, Canada

## Abstract

The interior of western Canada, like many similar cold mid- to high-latitude regions worldwide, is undergoing extensive and rapid climate and environmental change, which may accelerate in the coming decades. Understanding and predicting changes in coupled climate–land–hydrological systems are crucial to society, yet limited by lack of understanding of changes in cold region process responses and interactions, along with their representation in most current generation land surface and hydrological models. It is essential to consider the underlying processes and base predictive models on the proper physics, especially under conditions of non-stationarity where the past is no longer a reliable guide to the future and system trajectories can be unexpected. These challenges were forefront in the recently completed Changing Cold Regions Network (CCRN), which assembled and focused a wide range of multi-disciplinary expertise to improve the understanding, diagnosis, and prediction of change over the cold interior of western Canada. CCRN advanced knowledge of fundamental cold region ecological and hydrological processes through observation and experimentation across a network of highly instrumented



research basins and other sites. Significant efforts were made to improve the functionality and process
representation, based on this improved understanding, within the fine-scale Cold Regions Hydrological
Modelling (CRHM) platform and the large-scale Modélisation Environnementale Communautaire (MEC) –
Surface and Hydrology (MESH) model. These models were, and continue to be, applied under past and
projected future climates, and under current and expected future land and vegetation cover
configurations to diagnose historical change and predict possible future hydrological responses. This
second of two articles synthesizes the nature and understanding of cold region processes and Earth
system responses to future climate, as advanced by CCRN. These include changing precipitation and
moisture feedbacks to the atmosphere; altered snow regimes, changing balance of snowfall and rainfall,
and glacier loss; vegetation responses to climate and the loss of ecosystem resilience to wildfire and
disturbance; thawing permafrost and its influence on landscapes and hydrology; groundwater storage and
cycling, and its connections to surface water; and stream and river discharge as influenced by the various
drivers of hydrological change. Collective insights, expert elicitation, and model application are used to
provide a synthesis of this change over the CCRN region for the late-21st century.

## 1. Introduction and objective

The interior of western Canada is a region undergoing rapid, widespread, and severe hydro-climatic and
environmental change. This region is emblematic of the scientific and societal challenges in cold regions
around the world where snow, ice, and frozen soils dominate water cycling processes. Parts of western
and northern Canada have experienced some of the highest rates of climate warming anywhere in the
world (IPCC, 2013; Bush and Lemmen et al., 2019) and there have been systematic patterns of change in
climate regime and cryospheric response (DeBeer et al., 2016), including a shift in the phase of
precipitation ($P$) toward more rain and less snow, earlier snowmelt and decreasing extent, duration, and
maximum depth of seasonal snow cover, retreating glaciers, warming and thawing permafrost, declining
freshwater ice cover period, and an earlier spring freshet. Against this backdrop of change, western
Canada has been subjected to a series of recent, and in some instances record-breaking, extreme events
such as floods, droughts, and wildfires. Human interventions and land and water management have also
affected the environment and river systems, with infrastructure developments such as dams, diversions,
and irrigation networks, along with industrialization, agricultural development, and urbanization, thereby
altering natural ecosystems and water cycling. Future projections of warmer climate, altered $P$ phase and
patterns, and more extreme events (Bush and Lemmen, 2019; Stewart et al., 2019), together with
increasing human pressures, indicate that the region will continue to undergo rapid change to conditions
never before experienced, posing difficult management and decision-making challenges (e.g., Razavi et
al., 2020).
Improved understanding and prediction of the changes in coupled climate–land–hydrological systems are
crucial for managing land and water systems, and informing governance and policy direction here and in
other similar regions globally. The processes of change in cold regions are manifold and complex, and
there is significant uncertainty with the prediction of future change. Often, modelling and projections of
hydrological change are based on over-simplistic or empirical approaches and models that fail to
adequately capture the interconnected process drivers and responses. It is unclear to what extent the
model structures and parameterizations are valid under highly non-stationary conditions, and hence
whether the results are meaningful under future climates and land and vegetation cover states. There
has been much speculation about how cold regions will change, but, in many cases, this has not been
based on appropriate process understanding, which is itself limited.





These issues and challenges were forefront in the goals of the recently completed Changing Cold Regions Network (CCRN; 2013-18; www.ccrnetwork.ca), described by DeBeer et al. (2015; 2016) and Stewart et al. (2019). CCRN aimed to integrate existing and new sources of data with improved predictive and observational tools to understand, diagnose and predict interactions amongst the cryospheric, ecologic, hydrologic, and climatic components of the changing Earth system at multiple scales. Its specific geographic focus has been on the cold interior of western Canada, and in particular, the two major river systems of the region – the Saskatchewan and Mackenzie River Basins (Fig. 1). The overall science objectives of CCRN were to:

1. **Document and evaluate observed Earth system change**, including hydrological, ecological, cryospheric and atmospheric components over a range of scales from local observatories to biome and regional scales;
2. **Improve understanding and diagnosis of local-scale change** by developing new and integrative knowledge of Earth system processes, incorporating these processes into a suite of process-based integrative models, and using the models to better understand Earth system change;
3. **Improve large-scale atmospheric and hydrological models** for river basin-scale modelling and prediction to better account for the changing Earth system and its atmospheric feedbacks; and
4. **Analyze and predict regional and large-scale variability and change**, focusing on the governing factors for the observed trends and variability in large-scale aspects of the Earth system and their representation in current models, and the projections of regional scale effects of Earth system change on climate, land and water resources.

Key to the success of the network was the ability to observe and diagnose change across the region, and hence provide a platform of data (e.g., see https://essd.copernicus.org/articles/special_issue901.html) and scientific insights to inform model development and application for the analysis and prediction of change. A multiscale observatory was developed, based where possible on existing experimental sites with historical data records (Fig. 1), and this formed the heart of the program, enabling process responses and interactions to be monitored across the different ecological regions, and at the scales of small river basins and major river systems. In conjunction with the experimental and observational program, modelling research aimed at improving the capability of fine and large-scale models to represent key cold region processes, and to diagnose the complex and interacting factors underlying the observed changes over the CCRN region. Finally, these models have begun to be used, in conjunction with expert elicitation, to examine likely future system trajectories for the purposes of informing management and policy and addressing other stakeholder concerns. In doing so, CCRN assembled and focused a wide range and depth of multi-disciplinary expertise to address the network's aims and to develop insights into the process controls across the CCRN domain.

This article draws together the expert understanding and process insights from CCRN, together with modelling results at different scales, to examine the key drivers of change and to highlight the most likely anticipated future system trajectories across the interior of western Canada. This follows Part 1 (Stewart et al., 2019), which synthesized CCRN's collective assessments of future climate conditions and the associated seasonal patterns, along with particular $P$- and temperature-related phenomena. The specific objective of this second article is to illustrate how these changes in the climate system will manifest as changes in land and vegetation cover, cryospheric states, and hydrological cycling.

The article is organized as follows: Section 2 provides a brief overview of CCRN's geographic domain and the two major river basins. Section 3 examines a number of different cold region processes, their interactions and responses to climate, and their influence on water cycling. This highlights complexities that most Earth system models fail to capture. Section 4 briefly describes the advancements in fine-scale



and large-scale process-based hydrological models during CCRN, along with their application for the
diagnosis and prediction of change, while Section 5 provides a synthesis of this change over the CCRN
region for the 21st century. Section 6 provides concluding remarks and guidance for further research.

## 2. Ecological regions and river systems of the interior of western Canada

The interior of western Canada spans a wide range of climatic, ecological, and physiographic regions (Fig.
1), and has many of the physical attributes common to cold regions worldwide (Woo et al., 2008). This
includes extensive areas of permafrost and seasonally frozen ground, snow and ice cover through a large
part of the year, and water cycling that is driven largely by seasonal patterns of energy availability. The
principal river systems include the Saskatchewan and Mackenzie Rivers and their respective 406,000 km²
and 1.8 million km² drainage basins (Fig. 1). These encompass Prairie, Boreal (including Taiga), Tundra,
and Cordillera landscapes (CEC, 1997).
The Saskatchewan River originates in the Rocky Mountains of Alberta and Montana, and flows through
the province of Saskatchewan and into Manitoba, discharging into Lake Winnipeg. Most of the flow
originates in the mountains, which provide roughly 80% of total discharge (Pomeroy et al., 2005). The
basin is mostly situated within the Prairies, a key agricultural region, and Boreal Plain; the transition
between these ecological regions is dynamic and largely coincides with an annual water balance threshold
where $P$ equals potential evapotranspiration (PET), with a moisture surplus to the north and deficit to the
south (Ireson et al., 2015). In the southern and central portions of the basin, part of the Palliser Triangle,
the climate is among the most arid in Canada (Szeto, 2007). The landscape is mostly post-glacial
topography, with large numbers of small depressions, and poorly developed and internally drained stream
networks (Pomeroy et al., 2005; Martz et al., 2007). Approximately 40–50% of the basin does not
contribute to river flows, with large-scale connectivity only developing in exceptionally wet conditions,
and only a very small percentage (~1%) of the flow in the main river originates within Saskatchewan (Martz
et al., 2007). The Prairie climate leads to large variability in local water flows and storages, as for example
seen in the extreme drought of 1999–2004 (Hanesiak et al., 2011) and the high water levels and floods of
the following decade (Dumanski et al., 2015; Szeto et al., 2015). Numerous environmental, societal, and
management challenges exist in the Saskatchewan River Basin (Wheater and Gober, 2013; Gober and
Wheater, 2014), and the South Saskatchewan River has been described as Canada's most threatened river
(WWF, 2009). Irrigation is the dominant consumptive use of water, and despite Canada's reputation as
water-rich country, water resources are fully allocated in southern Alberta. Dam storage and hydropower
development have caused major changes in the seasonal flow regime, impacting the habitats of the
10,000 km² Saskatchewan Delta, located at the Saskatchewan–Manitoba border.
The Mackenzie River drains about 20% of the Canadian land mass, spanning parts of British Columbia,
Alberta, Saskatchewan, the Yukon and Northwest Territories, and is the single largest North American
source of freshwater to the Arctic Ocean (Stewart et al., 1998; Rouse et al., 2003; Woo et al., 2008; WWF,
2009). The Mackenzie River has a number of major tributary rivers, including the Athabasca, Peace, and
Liard Rivers, as well as other smaller tributaries; overall, mountainous western parts of the basin
collectively provide about 60% of total flow (Woo et al., 2008). There are three major deltas—the Peace–
Athabasca, the Slave, and the Mackenzie, which host diverse ecosystems. The basin covers large areas of
Boreal and Taiga Forest, with relatively low relief and underlain by glacial plains in the south and south-
west, and by the Precambrian Shield with slightly more undulating topography in the east (Woo and
Rouse, 2008). Much of the central and northern parts of the basin are underlain by discontinuous and
continuous permafrost, which is thawing at an accelerating rate (Burn and Kokelj, 2009; Baltzer et al.,





2014).  In the plains region, the basin includes several very large lakes, and a large portion of the area is
covered by smaller lakes and wetlands (Woo et al., 2008). Climate conditions are cool, with considerable
intra- and inter-annual variability in air temperature, and the region is a source area for cold, continental
air masses (Szeto et al., 2008). The basin is a globally important resource that affects the welfare of people
throughout the western hemisphere and globally, yet the ecological, hydrological, and climatological
regimes are changing rapidly and are threatened by global warming and human impacts (RIFWP, 2013).
While the majority of the river basin is largely undisturbed, local impacts on river flows and ecosystems
arise in the headwaters, due to operation of the Bennett Dam on the Peace River, and in downstream
areas, for instance, due to operations of the Athabasca oil sands.
Over this region, past changes in stream and river discharge have exhibited a trend towards earlier spring
freshet and river ice breakup and an increase in winter discharge in many northern basins (DeBeer et al.,
2016).  Other changes have included increasing importance of rainfall in generating flood events
(Dumanski et al., 2015; Burn and Whitfield, 2016) a shift in flood regime along the continuum from
snowmelt to more mixed and rainfall-driven regimes (Burn and Whitfield, 2018), and in spite of warming
spring air temperatures, delayed spring streamflow in some areas of the southern Arctic (Shi et al., 2015).
Naturalized flows (after accounting for the changes due to reservoir operations and water withdrawals)
of the South Saskatchewan River have exhibited a steady decline since the early-20th century, with late
summer volumes declining at a greater rate than the annual discharge (Pomeroy et al., 2009).  Flows in
the Mackenzie River since the early 1970s have shown a shift in timing of peak flows of several days, an
increase in maximum discharge of about 3,000 m³/s, and a rise in winter base flows (Yang et al., 2015).

## 3. Process interactions, changes, and their influence on water cycling

Field-based observations and experimentation across the network of WECC observatories (Fig. 1) and at
other sites has provided key insights on process interactions and responses.  Here we summarize these
insights for several important hydrological and ecological processes.

### 3.1 Precipitation recycling and evapotranspiration

*P* and evapotranspiration (ET) are important terms in the water cycle and even minor shifts in their relative
magnitudes can have critical impacts on surface water availability, streamflow, and groundwater storage.
Recent changes in *P* over western Canada have shown regional and seasonal variations, with annual and
winter increases in volume in the north, and more significant winter decreases in the southern interior
(Vincent et al., 2015; DeBeer et al., 2016). Pervasive warming has led to notable declines in the fraction
of winter *P* falling as snow (Vincent et al., 2015; Dumanski et al., 2015). Historical variations and patterns
of ET in western Canada have shown mixed trends, in part, due to the challenges with measurement, data
availability, and modelling of ET (Mortsch et al., 2015). ET is affected by many variables, including
precipitation, air temperature, surface and soil moisture availability, net radiation, wind speed, humidity,
and vegetation characteristics. Thus, it is spatially highly variable over heterogeneous landscapes, and is
sensitive to changing climate and to land cover change (Zha et al., 2010). Changes in *P* amount and
character are controlled to a considerable degree by global and continental-scale conditions and their
influence on regional circulation, air mass characteristics, and smaller scale variability (e.g., Stewart et al.,
2019).  Some further considerations of interactions at the surface and land–atmosphere feedbacks that
affect local *P* processes are discussed here.


Regional moisture recycling between $P$ and ET is prevalent and provides a significant portion of the warm-
season $P$ across much of the Saskatchewan and Mackenzie River Basins (Szeto, 2007; Szeto et al., 2008).
It represents an important mechanism of moisture transport, in some instances leading to intense rainfall
and flooding (Li et al., 2017), and may have an important role in sustaining wet (or dry) conditions on
seasonal to inter-annual time scales.  For example, there are important feedbacks between ET and $P$,
where an increase in $P$ is likely to increase ET, but the increase in $P$ itself could also be a result of increasing
land ET and stronger moisture recycling (Trenberth, 1999; Dirmeyer et al., 2009).  In future, under a
warming climate, earlier disappearance of the seasonal snow cover will act to increase regional ET in
spring as a result of the reduction in surface albedo, increase in net radiation to the ground surface,
increase in overall surface temperature, thaw of frozen ground, and increase in exposure of wet soils.
Shorter ice cover duration, especially in more northern lakes, will lead to increased lake evaporation and
will therefore also play an important role in providing local moisture sources to downwind regions.  These
effects, together with earlier onset of ET from vegetation as a result of changes in the timing of leaf
emergence, will enhance local atmospheric moisture supply in spring, possibly further enhancing the
projected increase in March-April-May $P$ (see Fig. 5 of Stewart et al., 2019).  Later freeze-up in the fall can
have similar effects, producing more lake effect snowfall, for example.
Kurkute et al. (2020) simulated future changes in $P$ and ET over the Saskatchewan and Mackenzie River
Basins using a pseudo-global warming (PGW) approach with a high resolution (4km) Weather Research
and Forecasting (WRF) model (Li et al., 2019).  Under the RCP8.5 radiative forcing scenario, their results
show increases in $P$, ET, and moisture recycling in both basins for the late-21st century (2085–2100)
relative to their control period (2000–2015), but with considerable seasonal and spatial variations (Fig. 2).
In the early spring (March and April), increases in $P$ are projected to exceed increases in ET leading to
increasing snowpacks and/or soil moisture, but by May, the earlier snowmelt and increased atmospheric
evaporative demand lead to greater increases in ET compared to $P$ and drying of soils over much of the
Prairies and Boreal Forest of western Canada (see Figs. 9–10 of Kurkute et al. (2020)).  This pattern
continues into summer, and by July and August, simulated future $P$ decreases in these parts of the region
(most of the Saskatchewan River Basin), due in part to the decrease in soil moisture and surface water
availability in the antecedent spring months.  Although there is a simulated increase in moisture recycling
in the warm season, the excess of ET over $P$ is associated with an increase in atmospheric moisture
divergence (i.e., transport out of the region).
Changes in ET also occur as a consequence of land cover and vegetation changes.  Vegetation cover in
turn is influenced by soil moisture (which is controlled by topographic position and surficial geology) but
also by disturbance and succession dynamics (Ireson et al., 2015).  The main vegetative controls on ET
include leaf and canopy characteristics (vegetation height, LAI, leaf shape, stomatal behaviour), and
rooting depth and dynamics (Zha et al., 2010; Black and Jassal, 2016; Nazarbakhsh et al., 2020). Margolis
and Ryan (1997) showed that, due to physiological limitations to transpiration in Boreal needleleaf trees,
they have much lower ET rates than deciduous species, even when soil water is abundant.  This is
consistent with observations at the Boreal Ecosystem Research and Monitoring Sites (BERMS) flux towers
(Fig. 1, site 7), showing a mature aspen stand with higher ET than a mature black spruce, which had higher
ET than a jack pine stand (Fig. 3). Kljun et al (2006) attributed these differences to a combination of type
of tree species, topography and soil type.  Very young forest stands have also been shown to have much
lower rates of ET than older stands (Granger and Pomeroy, 1997).  Thus, shifts in Boreal Forest
composition and structure, from coniferous to deciduous or mixed-wood, or from black spruce to jack
pine (discussed in Sect. 3.4 below), will have potentially large, but species-specific effects on regional ET.



In the northern parts of the CCRN region, thaw-induced landscape change (Sect. 3.5) and expansion of
shrubs (Sect. 3.4) are among the key drivers of changes in ET. Increasing thaw depth and shrinkage of
permafrost-underlain areas impact growth and physiological processes of the trees through drying of the
rooting zone, driving decreases in the productivity of black spruce-dominated sub-Arctic forests and
reduction of sap flow and ET (Patankar et al., 2015; Sniderhan and Baltzer, 2016). At the same time,
however, the conversion from forest to wetland associated with permafrost thaw acts to expand areas of
open, freely evaporating water surfaces, counteracting this effect (e.g., Carpino et al., 2018). Warren et
al. (2018) demonstrated that at Scotty Creek (Fig. 1, site 13), ET attributable to black spruce accounted for
less than 1% of landscape ET, suggesting areas of open water are of much greater importance to the water
balance regionally. The expansion of shrubs in northern tree line and Tundra environments will likely
increase regional ET in the snow-free period. For example, Zwieback et al. (2019b) found that rainfall
interception losses from birch shrubs at Trail Valley Creek in the southern Arctic (Fig. 1, site 11) reduced
below-canopy rainfall by 15–30%, but that losses depend on shrub species and density. Shrubs can
efficiently reduce stomatal conductance under conditions of high vapor pressure deficit and their shading
effect can act to limit surface evaporation under dense shrubs (Lund, 2018), further complicating the
responses to shrub expansion. Shrub–snow interactions (Sect. 3.2) essentially act to retain winter
snowfall and increase post-melt water availability, resulting in greater ET (Pomeroy et al., 2006; Ménard
et al., 2014).

## 3.2 Snow regime change and snow–vegetation interactions

Over western Canada during the past several decades there has been a widespread reduction in snow
depth, snow cover extent, and seasonal duration, with a shorter snow cover period of between one to
two months, mostly due to earlier melt in spring (Brown et al., 2010, 2020; Mudryk et al., 2018; Marsh et
al., 2019). Projected climate warming over the coming decades will continue to cause ubiquitous changes
in snow regime, including i) a greater fraction of $P$ in the form of rain as opposed to snowfall, especially
during shoulder seasons, at lower elevations, and in more southerly locations, ii) more frequent rain-on-
snow events, iii) warmer and wetter snowfall, iv) more mid-winter melt events as air temperature crosses
the freezing point more frequently, and v) earlier spring melt and snow cover depletion (Fig 4). This will
also cause distinct changes in runoff, with further transition from snowmelt to rainfall-dominated regimes.
The transitions from snowfall to rain and from snow-dominated to rain-dominated hydrological systems
are particularly sensitive where and when conditions are relatively warm and large amounts of $P$ occur
near 0°C (Mekis et al., 2020). For example, analysis by Harder and Pomeroy (2014) in the Rocky Mountain
Front Ranges at Marmot Creek (Fig. 1; site 2) showed that a significant proportion of the observed $P$
events, recorded as either snowfall or rain, occurred within just a few degrees plus or minus of 0°C as air
temperature or hydrometeor temperature. Even slight warming could lead to rain becoming dominant
at such locations. Shi et al. (2015) described the effects of increased rainfall during the snowmelt runoff
period at Trail Valley Creek.
Hillslope-scale snowmelt runoff is potentially highly vulnerable to warming temperatures and associated
changes in the amount and phase of precipitation. For instance, at three small (5 ha) hillslopes in the
Saskatchewan Prairie, Coles et al. (2017) found that increases in summer rains were buffered by the
unfrozen, deep, high-infiltrability soils. In contrast, winter and spring melt onto frozen ground with limited
soil infiltrability resulted in runoff responses that more closely mirrored the snowfall and snowmelt
trends. Increasing occurrence of mid-winter melt events can also alter the timing and magnitude of
depression-focused groundwater recharge (Pavlovskii et al., 2019) and may lead to more basal ice
formation, producing complex runoff responses in spring. Follow-on hillslope-scale analysis by Coles and



McDonnell (2018) found evidence for filling of micro- and meso-depressions on the slope, followed by
macro-scale, whole-slope spilling.  While surface topography is relatively unimportant under unfrozen
conditions on low relief and high infiltrability Prairie sites, surface topography was of critical importance
for connectivity and runoff generation when the ground was frozen during the brief, annual snowmelt
pulse.  Under climate warming, losing this brief period of surface topographic control on runoff generation
could have large implications for hillslope runoff, depending on basal ice formation, among other factors.
Warming can also lead to other important, and sometimes unanticipated, responses in snow
accumulation, redistribution, and ablation processes (Fig. 4). Earlier onset of spring melt of the seasonal
snow cover shifts snowmelt timing to conditions of lower incoming solar radiation (Pavlovskii et al., 2019).
Paradoxically, this can lead to a reduction in daily and seasonal average ablation rates and a longer overall
period of melt (Pomeroy et al., 2015; Musselman et al. 2017) in some cases, but not in the Arctic where
earlier and faster melts are predicted (Krogh and Pomeroy, 2019).  This is counterintuitive and would not
be captured by simple temperature-index melt models (Pomeroy et al., 2015).  Warmer and wetter snow
has lower susceptibility to wind transport (Li and Pomeroy, 1997), leading to a potential reduction in
blowing snow transport and sublimation losses, which can partially offset reductions in snow water
equivalent (SWE) due to direct effects of climate warming.  Model results by Pomeroy et al. (2015) at
Marmot Creek indicate the reduction of blowing snow transport and sublimation with warming of up to
5°C reduces the redistribution by transport by up to 50% and losses from sublimation by up to about 30%.
This would also have important, but at present, poorly understood consequences on the redistribution of
snow, the variability and patterns of SWE over the landscape, and the timing and rate of snow cover
depletion (e.g., DeBeer and Pomeroy, 2017).  Suppression of blowing snow would lead to a more uniform
spatial distribution and thus more rapid decline of snow-covered area that could not be compensated for
by the variability in melt energy (Schirmer and Pomeroy, 2020).
Snow–vegetation interactions further affect hydrological responses, and the impacts of vegetation change
can equal or exceed those due to climate alone (Rasouli et al., 2019).  A conceptual summary is shown in
Fig. 4.  With rising temperatures, warmer and wetter intercepted snow is more likely to fall to the ground
instead of remaining in the forest canopy, where it would otherwise mostly sublimate.  Snowfall
interception efficiency is relatively insensitive to air temperature (Hedstrom and Pomeroy, 1998) and thus
warming is unlikely to lead to large changes in initial interception amounts.  But retention of the
intercepted snow load is highly temperature dependent (Ellis et al., 2010) and so warming promotes faster
unloading and a lower sublimation loss.  This acts in combination with reduced wind transport of snow on
the ground to offset reductions in SWE due to direct warming effects (Pomeroy et al., 2015).  Forest
canopy structure, density, and species composition also significantly influence interception loss.  Thinning
of existing forest cover, reduction in leaf area index (LAI), and transition from coniferous to deciduous
species, which are expected as a result of increasing human and natural disturbance and wildfire (Sect.
3.4), will lead to greater surface snow accumulation due to the reduction in canopy interception and
sublimation, but at the same time will expose more of the snow surface to increasing net radiation and
an accompanying increase in ablation rates.
In open, windswept environments dominated by short vegetation such as grasses, crops, and shrubs,
expansion across the landscape and/or increasing height and density of vegetation influences surface
water availability and land–atmosphere energy and moisture exchanges.  Shrub expansion acts to
enhance local snow accumulation through more trapping of wind-blown snow and suppression of blowing
snow redistribution and sublimation (Pomeroy et al., 2006; Ménard et al., 2014; Wallace and Baltzer,
2019).  Shrubs reduce albedo in the spring but are buried in winter and have little effect on albedo in
summer.  Their canopy reduces latent heat fluxes from snow in the spring and initially accelerates melt



when partly exposed and then retards snowmelt when the shrub canopy is fully exposed (Pomeroy et al.,
2006; Wilcox et al., 2019).  Increasing crop stubble height acts to retain more snow, and to increase melt
rates, infiltration, and meltwater runoff (Harder et al., 2019).
### 3.3 Glacier loss
In western Canada and globally, glaciers have been predominantly losing mass and retreating in extent,
with an apparent acceleration in their wastage in recent decades (Demuth and Ednie, 2016; Menounos et
al., 2019; DeBeer et al., 2020).  Even in the absence of further warming, many of these glaciers are out of
balance with the current climate, given their present configuration (Marzeion et al., 2018).  This indicates
that they will further recede to adjust their geometry to the current climate, with a typical response time
of several decades for glaciers in western Canada (Marshall et al., 2011; Marzeion et al., 2018).  Ongoing
climate change is expected to further exacerbate the current imbalance and lead to additional retreat
(Clarke et al., 2015).
Mass balance (the net gain or loss of snow and ice averaged over the glacier surface) responds directly to
climate perturbations, whereas glacier extent, form, and flow patterns exhibit delayed and modified
responses to mass balance changes (e.g., Clarke et al., 2015).  Glacier responses are also influenced by
secondary factors such as temperature effects on ice flow and meltwater availability at the glacier bed,
which affects glacier sliding.  In general, warmer air temperatures lead to greater specific ablation rates
and a longer melt season, and may reduce accumulation depending on the area–elevation distribution of
individual glaciers and the nature of $P$ changes.  Many glaciers and icefields in the CCRN region receive
snowfall year round at high elevations and some rainfall in the summer.  With climate warming, the
proportion of rainfall events increases and the late summer snowline moves to higher reaches of the
glaciers, exposing firn and bare ice, which melt faster than snow due to their lower albedo.  Dust,
impurities, and algae in the snow and ice become more concentrated on glacier surfaces as a consequence
of high melt rates, in turn reducing the albedo and further enhancing melt (Williamson et al., 2019; DeBeer
et al., 2020).  There may also be an interaction with wildfire in western Canada, with deposition of black
carbon and forest-fire fallout further reducing glacier albedo and providing nutrients to microbial
communities (e.g., Marshall and Miller, 2020).  High thinning rates in the upper accumulation area of many
glaciers in western Canada indicate that these processes are well under way (Pelto et al., 2019), while
reductions in accumulation zone extent can lead to rapid glacier disintegration, and even complete
disappearance.  Glacier fragmentation and detachment of tributary ice streams leads to loss of ice supply
to lower reaches, which can then become stagnant and melt out.
There are other important glacier–climate feedbacks.  Energy balance conditions shift in response to
glacier retreat; for example, ice-free marginal areas and valley walls contribute turbulent energy supply
and longwave radiation fluxes to the glacier, and these fluxes can be enhanced as glaciers thin and retreat,
increasing ablation rates.  The presence of glacial ice helps to regulate local climates and preserve cold
conditions.  As reduced snow accumulation leads to a reduction in glacier mass balance, so a reduction in
glacier extent leads to a reduction in snow accumulation, given that the glacier surface, which is ≤ 0°C,
helps retain snow cover (Marshall et al., 2011).
Projections of future glacier change indicate that glaciers in the Rocky Mountains will lose roughly half
their total area and volume by mid-century, and as much as 90% or more by the end of the 21[st] century
under a 'business as usual' (RCP8.5) climate scenario (Clarke et al., 2015).  By mid-century, many valley
glaciers will have retreated substantially up-valley, and by late in the century even high elevation glaciers



and icefield plateaus will be greatly reduced or will have disappeared entirely (Fig. 5). Even the Columbia
Icefield, the largest and among the highest elevation ice masses in the Rocky Mountains, is projected to
disintegrate into several small vestigial patches of ice near the tops of the highest peaks by the late-21st
century. There are not comparable studies for the glaciated regions of the Mackenzie Mountains,
Northwest Territories, but the observed patterns of recent change are similar to glaciers in the Rockies
and the future change is expected to be similar.
From a hydrological perspective, as glacier loss progresses, glacier wastage contributions and enhanced
ablation will increase glacial contributions to discharge towards "peak water" (Huss and Hock, 2018),
followed by a decline in glacier runoff due to loss of ice-covered area, even with further warming and
increasing specific ablation rates. It remains uncertain where, when, and over what scales this will occur,
although some studies have indicated that peak water has already passed in parts of southwestern Canada
(Moore et al., 2020). Clarke et al. (2015) projected that peak runoff of glacier meltwater will occur
between 2020 and 2040. Projections of glacier decline on the eastern slopes of the Rocky Mountains by
Marshall et al. (2011) indicate a substantial decline in glacier contributions to discharge from about 1.1
km$^3$ per year early this century to 0.1 km$^3$ per year by late in the 21st century. With the loss of glaciers,
the buffering effect that glacial storage can provide for discharge variations (e.g., during drought years) in
the mountain headwaters will become increasingly diminished.

## 3.4 Northern vegetation, wildfire, and loss of ecological resilience

Ecosystem change can have profound effects on hydrological response and land–atmosphere feedbacks,
yet the complexity of expected change and the associated uncertainty are often overlooked in
hydrological projections. Across the CCRN region, contemporary climate change is already having direct
impacts on northern ecosystems, defined here as including the southern Boreal Forest and its transition
with the Prairies, and the Cordillera. The interior of western Canada has been identified as a region of
maximum ecological sensitivity (Bergengren et al., 2011). Forests in the southern Boreal region of western
Canada have shown signs of declining productivity and increasing mortality associated with drought stress
or insect disturbances, including widespread dieback and mortality of aspen (Hogg et al., 2008), stand
fragmentation, and increases in tree mortality of up to 2.5% per year (Peng et al., 2011). Farther north,
remote sensing indices of vegetation greenness indicate that substantial areas of Tundra and northern
Boreal Forest have been increasing in vegetation productivity (Ju and Masek, 2016; Keenan and Riley,
2018; Sulla-Menashe et al., 2018). This is largely due to expansion of woody shrubs, such as alders and
tall willows (Myers-Smith et al., 2011, 2019; Lantz et al., 2013), infilling of forests near the northern tree
line (Lantz et al., 2019), and increases in tree growth rates (Sniderhan et al., 2020). Advancement of the
Taiga–Tundra tree line in response to recent trends of climate warming has been more variable (Harsch
et al., 2009; Dearborn and Danby 2018). Lantz et al. (2019) showed infilling of forests below tree line in
the Northwest Territories, but no increase in tree density above tree line in the Tundra. To the south in
the Rocky Mountains, Trant et al. (2020) observed widespread upward advance in alpine tree lines and
increases in tree density, with changes in growth form from krummholz to erect tree form.
Climate change alters terrestrial ecosystems broadly through changes to: 1) composition (vegetation,
soils, and wildlife), 2) configuration and disturbance patterns, and 3) function. This includes structural
changes to the current vegetation (above- and below-ground biomass, plant density, canopy height, LAI,
and rooting depth); changes to land cover distribution patterns (resulting from changes in the disturbance
regime and changes in competition, colonization, ecosystem resilience and vegetation succession
following disturbance); and functional changes (surface albedo, snow accumulation and melt, soil freeze





and thaw, ET, ecosystem productivity, decomposition, biogeochemical cycling, and wildlife habitat). The
direct climatic drivers of vegetation change include rising atmospheric $CO_2$ concentrations and
temperature- and moisture-induced shifts in plant community function and vegetation distributions.
However, over the 21[st] century the greatest impacts of climate change on vegetation dynamics are
expected to be indirect, via increased frequency and intensity of disturbance (wildfire, insect outbreaks,
and other landscape-scale disturbances; Turetsky et al., 2017) leading to losses of ecosystem resilience.
These intensified disturbance processes can cause ecosystems to reach critical tipping points, triggering
ecological state change (reviewed by Johnstone et al., 2016). Imposed on the climate-induced changes in
vegetation will be the potential for changing human activities (e.g., logging, land-clearing for agriculture
and mining; Landhausser et al., 2010; Hannah et al., 2020), some of which will interact with climate change
to accelerate vegetation change.
Northern ecosystems are expected to be most resilient to disturbances and environmental conditions that
are within the historic range of variability and previous adaptation (Keane et al., 2009; Johnstone et al.,
2016; Seidl et al., 2016). Many northern ecosystems may be initially resistant to change, because
feedbacks associated with long-lived vegetation help to maintain environmental conditions and ecological
functions that support ecological stability, even during directional environmental change (Chapin et al.,
2004). While fire has been a foundational process in the functioning and ecology of the Boreal Forest for
more than 5,000 years, an increase in the frequency of high-intensity fires, coupled with a warming
climate, may weaken ecosystem resilience and disrupt the historically stable cycles of forest succession.
The result may be a regime shift from one plant community to another and from one stability domain to
another (Johnstone et al., 2010c; 2016). Wildfire activity has increased in recent decades across the Boreal
Forest (Hanes et al., 2019) and there are indications that fires are burning more severely (Turetsky et al.,
2011) and deeper into stored legacy carbon (Walker et al., 2019), creating novel conditions for forest
regeneration (Johnstone et al., 2010a; Pinno et al., 2013). For example, stands may burn at young ages
before trees are old enough to generate seeds; these events, especially when they occur in combination
with unusually dry or warm years, can trigger regeneration failures and cause shifts to non-forested states
(Brown and Johnstone, 2012; Whitman et al., 2018). Stand-replacing wildfires initiate new phases of
forest regeneration where seedlings may be much more sensitive to climate conditions than in an
established stand where canopy trees substantially alter the local microclimate (Johnstone et al., 2010b;
Davis et al., 2019; Hart et al., 2019). There is consensus that in northern forests, fire frequency and
severity will continue to increase (Rogers et al., 2020).
Projections of future wildfire-induced ecosystem change in the Boreal Forest are challenging and highly
uncertain. Increasing fire will result in a younger forest, widespread replacement of black spruce stands,
and higher proportions of deciduous broadleaf species or jack pine (e.g., Johnstone et al., 2010a), with
greater change in the south than the north. CCRN developed a plausible scenario of post-fire replacement
of evergreen needleleaf forest (ENF) with deciduous broadleaf forest (DBF) across the Boreal Forest,
as described in the Appendix, for the purpose of use in hydrological model future projections (Fig. 6).
Although this is simply a scenario, and not a projection with an associated confidence level, the resulting
forest change due to increasing wildfire is potentially great. For both the mid and late-century periods,
there is a considerable reduction in DBF across the southern parts of the Boreal Plain, as a result of
increasing fire and the conversion of forest to grassland. Farther north and west, in the Taiga Plain, the
Shield, and the Western Cordillera, there is extensive and progressive replacement of ENF with DBF as a
result of both climate and fire-driven changes in forest succession. In reality, DBF and jack pine stands
tend to be more resilient to fire (Hart et al., 2019), and less flammable in the case of DBF, and so their
expansion may partially counter the increase in fire occurrence expected under a warmer climate.





Insects represent another form of disturbance with high potential for disrupting forest successional
patterns, and may also lead to the replacement of black spruce stands by mixed-wood and deciduous
species (Pureswaran et al., 2015). Forest insects may expand northwards if warmer winter temperatures
increase potential rates of population growth (Post et al., 2009; Bentz et al., 2010). For the first time, pest
populations of mountain pine beetle have been found in the Northwest Territories (GNWT, 2013).
Likewise, unusual outbreaks of spruce bark beetle in the Yukon and Alaska have been associated with
warm winter temperatures that allow increased insect survival through the winter (Berg et al., 2006). In
some cases, forests have exhibited high levels of resilience to new disturbance conditions, as in the rapid
recovery to bark beetle outbreaks in the southwest Yukon (Campbell et al., 2019).
Across the northern and alpine tree line and tundra areas, displacement of shrubs by ENF and larch forest
will occur in areas where sparse forest cover exists (e.g., Mamet et al., 2019), while above the tree lines,
shrub expansion into tundra environments will likely continue with warmer temperatures and increasing
water availability. Large shifts in tree line position are not expected over the 21$^{st}$ century due to both
biological and geological constraints. At the northern tree line, the limited reproductive capacity of the
tree species results in low seed availability, which restricts the rate of tree expansion into tundra
ecosystems (Brown et al., 2019; Harsch et al., 2009), although this is dependent on the nature of the tree
line as expanded upon in Harsch et al. (2009). Similarly, the advance of the alpine tree line is restricted
by geological and geomorphological controls such as avalanching, soil limitations, slope configurations
that generate harsh winds, and other seed establishment and growth-limiting factors (Macias-Fauria and
Johnson, 2013; Davis and Gedalof, 2018). Northern and montane shrub tundra areas will expand and
continue the greening trend, with conversion of dwarf-shrub and graminoid-dominated tundra to tall-
shrub tundra, resulting in more and taller shrubs, and an increase in LAI for existing patches. At fine scales,
the rate and location of shrub expansion are very heterogeneous due to combined moisture and nutrient-
driven responses (Wallace and Baltzer, 2019). For instance, although most infilling and recruitment is
expected to occur in valley bottoms, low-lying areas, and other locations with sufficient water availability,
excess moisture can carry nutrients downslope. Shrub Tundra is also susceptible to disturbance-induced
changes. Large fires can occur in Tundra environments (Mack et al., 2011), and increased fire activity may
occur if temperatures cross climate thresholds that have regulated fire activity in the past (Young et al.,
2017) or as fuel accumulates due to shrub expansion. Permafrost thaw also affects shrub colonization
(see Sect. 3.5). Shrub expansion can have multi-directional hydrological impacts (Grunberg et al., 2020),
including shrub–snow interactions (Sect. 3.2) and increasing ET (Sect. 3.1), warmer soils, greater thaw
depth, and thermokarst and subsidence, altering supra-permafrost layer storage, flow paths, and lake
development (Sect. 3.5).
In addition to the forest cover change scenario, CCRN developed a plausible scenario of 21$^{st}$ century shrub
expansion into tundra, grassland, and barren areas, described in the Appendix and shown in Fig. 7. While
there is uncertainty and this does not represent a confident projection, prolific shrub growth over the
Boreal and Taiga Cordillera, the Southern Arctic, and the Taiga Shield ecological regions is expected. The
gradual expansion northward is evident through the increase in shrub cover along the northern part of
the Mackenzie River basin and the movement of this growth zone to higher latitudes later in the century.
## 3.5 Permafrost thaw as a driver of landscape change and hydrologic rerouting
Climate warming has led to warming and increased thaw depth of permafrost across northern Canada
(Smith, 2011), with associated changes in characteristics of seasonally-frozen soils (e.g., timing of freezing
and thawing, frequency of freeze-thaw cycles, depth of frost, etc.). In southerly locations where



permafrost is discontinuous, shallow, and relatively warm (i.e., at or near the freezing point depression),
there has been widespread thawing and degradation of permafrost, with increasing supra-permafrost
layer thickness—including both the active layer (seasonally frozen) and the talik (perennially thawed)
(Connon et al., 2018). As a result of warming and shallower re-freeze depths during winter, active layer
thickness has been decreasing.  Where ice-rich soils occur, there has been active thermokarst
development, slumping, and ground surface subsidence (Olefeldt et al. 2016, Turetsky et al. 2019). In
permafrost lowlands of the Taiga Plain, soil thawing has led to subsidence and inundation of ground
surfaces resulting in extensive forest loss, fragmentation, and concomitant wetland expansion and
conversion mostly to sphagnum-dominated bogs (Baltzer et al., 2014; Helbig et al., 2016). In the southern
Arctic, increased permafrost thawing is leading to changes in channel permafrost conditions, increasing
winter groundwater flow in the channel, and increasing occurrence of aufeis formation (Ensom et al.,
12  2020).

Many northern ecosystems are underlain by ice-rich permafrost that is highly sensitive to thawing during
warm summers (Segal et al., 2016; Lewkowicz and Way, 2019) or following other disturbances (Williams
et al., 2013). Wildfire and combustion of insulating moss and peat layers affects permafrost temperatures
and can trigger thaw (Holloway et al., 2020).  The lateral expansion of thermokarst features increases
following wildfire activity; for example, Gibson et al. (2018) found that wildfire was estimated to be
responsible for 30% of permafrost thaw expansion in the southern Northwest Territories.  Some of the
energy driving the thaw of permafrost enters the permafrost bodies laterally from adjacent permafrost-
free terrains (Kurylyk et al. 2016).  As such, the rate of permafrost thaw and forest loss is accelerating as
patches of permafrost plateau become more fragmented, leading to greater proportional plateau edge to
total plateau area (Quinton and Baltzer, 2013; Baltzer et al., 2014; Carpino et al., 2018).  Reduced soil
stability during thaw events can cause substantial mass wasting through thermokarst and retrogressive
thaw slumps, with impacts on local vegetation and downstream drainage (Schuur and Mack, 2018). Once
the vegetation is disturbed, colonization by tall shrubs can cause a persistent change in vegetation state
due to altered patterns of snow accumulation and soil temperatures (Lantz et al., 2009; Schuur and Mack,
28  2018).

Quinton et al. (2009) proposed a conceptual model of canopy thinning and permafrost thaw in which
canopy thinning due to fire, disease, or other disturbance allows for an increase in local solar energy input
and leads to preferential ground thaw (Fig. 8). A local depression forms in the relatively impermeable
frost table and underlying permafrost table.  Such thaw depressions introduce a hydraulic gradient that
directs subsurface flow towards them so that thaw depressions soon become local areas of elevated soil
moisture content.  Since the thermal conductivity of wet soil is far more than that of dry soil, the vertical
conduction of energy to the thaw depressions increases due to the increased moisture content, and as a
result, a positive feedback is initiated which accelerates the thaw of the disturbed areas.  Wet conditions
prevent trees from re-establishing and a new, isolated flat bog is formed.  Many areas within the Taiga
Plain are highly susceptible to thaw through this process (e.g., Gibson et al., 2020) and widespread
replacement of forest-covered peat plateaus by wetlands is expected over the coming decades. A caveat
is that these ecosystems represent some of the strongest ecosystem-protected permafrost, so
undoubtedly a portion of permafrost peatland will linger, but this will depend on the degree of warming
and also fire (Stralberg et al., 2020).
The loss of permafrost is impacting water cycling across the northern parts of the CCRN region.  Land
surface subsidence and the collapse of peat plateaus to wetlands in the Taiga Plain alters drainage
networks, surface and groundwater storage distribution, and the transit of water across the landscape
(Fig. 8; Connon et al., 2014; 2018; Haynes et al., 2018; Quinton et al., 2019).  This incorporates individual



wetlands into the runoff contributing area, which expands deep into the interior of extensive plateau–wetland complexes as hydrological connections form between wetlands. The process results in both transient increases to basin discharge through the dewatering of incorporated wetlands, and longer-term increases in discharge arising from an expanded contributing area (Quinton et al., 2019). Another mechanism by which thaw influences runoff processes is by opening previously inaccessible subsurface flow pathways. Talik expansion provides an additional drainage path for wetland dewatering—one that conducts water throughout the year (Connon et al., 2018; Devoie et al., 2019). While this may give rise to transient increases in basin discharge due to the increased connectivity and dewatering of wetlands (Quinton et al., 2019), the process is not sustainable and may result in eventual drying of the landscape with increasing ET (Stone et al., 2019). Regeneration of black spruce forest may ultimately occur in the absence of permafrost, as has been observed further south Northwest Territories–British Columbia border (Carpino et al., 2018). In the Taiga Shield landscape, lake storage state can rapidly change the contributing area for runoff downstream and the landscape has a distinct threshold–response runoff regime (Ali et al., 2013). Wetlands are important "switches" in controlling the state of hydrological connectivity in the watersheds (Spence and Phillips, 2015). Permafrost thaw (and ultimately disappearance) may significantly affect this functioning, but it is unclear at what fraction of thaw progression major hydrological changes will occur.

### 3.6 Groundwater interactions and Prairie wetland processes

Over much of the Prairies and the Boreal Plain, groundwater discharge from shallow sand and gravel aquifers sustains year-round base flow in some small streams and can be an important component of the water balance of wetlands and of some lakes. Groundwater is thus important with respect to local water resources and in maintaining surface hydrological connectivity and ecosystem function. Groundwater provides rural water supplies and in some cases municipal supplies (Peach and Wheater, 2014), and whilst it is not used as a major source for irrigation water outside of the south-central parts of Manitoba, an issue facing some parts of the Prairies is the increasing reliance on groundwater as water demand rises and surface water becomes over-allocated (Council of Canadian Academies, 2009). Regional-scale groundwater depletion is not common in Canada, unlike other parts of North America (Rodell et al., 2018), but there have been numerous examples of isolated, human induced local-scale depletion in Alberta (e.g., Munroe, 2015). The water-table records in shallow (< 20 m) observation wells in the Prairie region show regular seasonal variations, with rises in spring and declines through the rest of the year. There have been no large long-term changes during 1960–2000, a noticeable drop during the 2000–2004 drought, followed by a rise in the following decade (Hanesiak et al., 2011). There are very few long-term observation wells in the Boreal Plain, but the detailed records of water table variations at the BERMS (Site 7, Fig. 1), together with hydrometeorological records, demonstrate the responses of the water table to changes in net water input to the subsurface throughout dry and wet periods and in various typical settings including peatlands and dry uplands (Anochikwa et al., 2012, Barr et al., 2012).

In the Prairies and Boreal Plain, lateral groundwater flow is slow due to the relatively flat terrain and the low permeability of the clay-rich glacial sediments underlying most of the landscape. As a result, subsurface water movement is mostly vertical—downward with infiltration, upward by root uptake—and the soil water and groundwater form a hydrological continuum. Rises of the water table are primarily driven by snowmelt infiltration and by focused recharge beneath ephemeral ponds in small wetlands and depressions that dry out within days or weeks after filling with snowmelt runoff (Bam et al., 2020). Recharge processes are sensitive to changes in snow accumulation, redistribution, and ablation processes (Sect. 3.2), and to land-use conversion (e.g., native grassland to cultivated fields, change in tillage


practice), which influences soil hydraulic properties and snowmelt infiltration and runoff (van der Kamp
et al., 2003). Most summer $P$ infiltrates only to the root zone and is taken up by vegetation, driving a
seasonal decline of the water table (Hayashi et al., 2016). However, summer infiltration can lead to rises
of the water table where it is near the ground surface, as in wetlands. As a result, the dynamics of the
shallow groundwater table are strongly controlled by the balance between infiltration and ET in response
to weather, vegetation, and seasons. It is also sensitive to inter-annual and inter-decadal fluctuations in
$P$ (Hayashi and Farrow, 2014). The water table in the Prairie and Boreal regions can fluctuate quickly but
is generally limited in range. When the water table rises near the ground surface, ET is increased and
lateral groundwater flow to surface waters becomes important within the highly fractured near-surface
materials (Hayashi et al., 2016; Brannen et al., 2015). This causes the water table to decline and provides
the negative feedbacks to limit the range of water table fluctuations to a few meters.
Groundwater processes are closely linked to the water regime (i.e., hydroperiod) of wetlands. Prairie
wetlands occur in the form of shallow marshes ("sloughs" or "potholes") with little accumulation of
organic matter, whereas Boreal wetlands primarily occur as peatlands. The spatial transition from Prairie
marshes to Boreal peatlands is coincident with the transitional ecotone between the Prairie and Boreal
Plain regions, described in Sect. 2 (see Ireson et al., 2015). The hydroperiod of prairie wetlands is
essentially controlled by a balance between water inputs from snowmelt runoff and $P$, versus ET losses
and sporadic overflow in wet periods (Hayashi et al., 2016). Groundwater outflow from these wetlands
due to the ET in the riparian zone also has a strong influence on the hydroperiod. Long-term (50+ years)
data collected at the St. Denis WECC observatory (Fig. 1, site 8) have demonstrated the dominance of
precipitation amounts in controlling the multi-decadal scale variability in hydroperiod (Hanesiak et al.,
2011; Hayashi et al., 2016).
The hydrology of Boreal peatlands has not been studied as extensively as that of Prairie wetlands, but
studies of a fen in the BERMS (Barr et al., 2012) have shown that it has a large water storage capacity and
supplies base flow to streams and to support the shallow water table in surrounding uplands during dry
periods. In contrast, the fen sheds water quickly to streams during wet periods when the water table rises
above the peat surface. Long-term studies in northern Alberta have shown that the type of glacial
sediments has a large influence on the groundwater exchange and runoff generation from the peatlands
(e.g., Devito et al., 2017).
Groundwater replenishment to deeper aquifers is restricted by the low permeability of overlying layers of
clay, clay-rich glacial till, and shale, and by the position of the aquifers within larger regional groundwater
flow systems (Cummings et al., 2012). In the Prairies, replenishment rates to confined aquifers generally
range from a few mm to a few tens of mm per year (van der Kamp and Hayashi, 1998). Recharge to the
water table represents a residual in the water balance and is highly sensitive to changes in the balance
between $P$ and ET; however, replenishment to deep aquifers is not sensitive to variations of the water
table and therefore responds slowly to climate change.
In the Western Cordillera the interaction of groundwater with surface waters is in many ways different
from the groundwater dynamics in the Boreal Plain and the Prairies. Groundwater plays an essential role
in sustaining base flow in the mountain headwaters of large river systems (Paznekas and Hayashi, 2016),
and may be of growing importance under climate change. Above the tree line in the Rocky Mountains,
primary aquifers are sedimentary landforms such as talus, moraine, and rock glacier (Hood and Hayashi,
2015; Harrington et al., 2018; Hayashi, 2020; Christensen et al., 2020), except in areas with substantial
karst systems. Groundwater storage in these landforms is relatively small compared to the SWE contained
in the seasonal snow cover (Hood and Hayashi, 2015), and groundwater discharge exhibits a fast recession



after snowmelt or rainfall events. However, this is generally followed by a slower recession and the
remaining storage allows these aquifers to sustain stable base flow during the rest of the year when there
is little recharge (Hayashi, 2020). The high topographic relief, together with significant heterogeneity in
bedrock and surficial deposits, influences patterns of vertical and lateral groundwater flow and recharge
and discharge processes. At lower elevations, aquifers include glacial and alluvial deposits of highly
permeable sands and gravels that drape mountainsides and underlie valley bottoms, usually 10s to 100 m
thick, but in some instances up to several hundred meters in thickness (Toop and de la Cruz, 2002). These
store larger quantities of water and provide a reliable supply for municipal and industrial uses. In
floodplain areas, the water table is usually near the ground surface and fluctuates with river levels.
Although mountain aquifers are able to buffer base flow against climate warming and associated changes
in surface water availability (e.g., Paznekas and Hayashi, 2016), anecdotal evidence has indicated that they
cannot sustain high flows in drought years, such as in 2015 when the spring–summer discharge of the Bow
River fell to about half its median rate at Banff, and to less than 10% at its mouth.

## 4. Process-based modelling of change in CCRN

Due to the complexity in process responses to climate and anthropogenic change in the CCRN domain and
other cold regions, there is significant uncertainty associated with model projections of future
hydrological change. While all models have limitations, detailed process-based models can yield
important insights into interactions and feedbacks, and large-scale models can be used with careful
selection of possible scenarios to quantify likely effects of future change. Here we describe CCRN's efforts
to improve model process representation, diagnose past change, and predict future change.

### 4.1 Fine-scale diagnostic and predictive modelling

Based on field studies and understanding from the WECC observatories, efforts were directed primarily
at improving functionality and expanding the capability of handling complex cold region processes within
CRHM (Pomeroy et al., 2007; www.usask.ca/hydrology/CRHM.php). CRHM is a flexible modelling system
that can be used to generate a process hydrology model, specific to the needs of the user and to the
availability of driving meteorological data and of basin biophysical information to select parameters. A
functioning model is built by selecting various process modules from a library; the modules incorporate
algorithms or sub-models that are based on several decades of hydrological research. Process algorithms
cover a wide range of phenomena specific to cold regions hydrology, which are then linked together to
represent specific elements of the hydrological system and cycling over distinct landscape units termed
"hydrological response units" (HRUs). Process studies and model developments focused on blowing snow
transport and sublimation over complex terrain (Aksamit and Pomeroy, 2018, 2020); snowmelt in
disturbed forests and on slopes; water flow through snowpacks (Leroux and Pomeroy, 2017, 2019); glacier
snow, firn and ice melt (Samimi and Marshall, 2017; Marshall and Miller, 2020; Pradhananga, 2020); snow
avalanching; soil moisture and hydraulic conductivity (Zwieback et al., 2019a); and freezing and thawing
of soils (Krogh et al., 2017; Williamson et al., 2018; Rowlandson et al., 2018; Lara et al., 2020).
CRHM was applied at a number of the WECC observatories as well as other sites in western North America
and run for historical periods using local meteorological observations, ERA-Interim (Dee et al., 2011),
and/or bias-corrected WATCH (http://www.eu-watch.org/) forcing data. It was verified using field
observations and then used to diagnose hydrological function of these basins, and predict and diagnose
historical change, such as the impact of changing climate, wetland drainage, glacier shrinkage and ice
exposure, permafrost thaw, and shrub growth/expansion on hydrological processes, cycling, and
streamflow hydrographs. It has also been run for late 21st century climates, downscaled using statistical
and dynamical methods. Future sensitivity and change was examined by perturbing climate forcing using
high resolution WRF modelled pseudo global warming under RCP8.5 (see Krogh and Pomeroy, 2019) or
using results from the North American Regional Climate Change Assessment Program (NARCCAP)
consisting of 11 regional climate models driven by outputs from multiple global climate models (GCMs)
for the SRES A2 emission scenario (see Rasouli et al., 2019). Hydrological responses to changing
vegetation, soils, and land cover were examined using current and expected future states of the basins.

## 4.2 Large-scale river basin modelling

CCRN worked with partners in Environment and Climate Change Canada (ECCC) to advance the
Modélisation Environnementale Communautaire (MEC) – Surface and Hydrology (MESH) model. MESH is a
stand-alone land-surface–hydrology scheme designed for both forecasting and open loop simulations
(Pietroniro et al., 2007). It uses a "grouped response unit" (GRU) approach to represent spatial
heterogeneity for parameter identification, with CLASS as the surface water and energy budget simulation
model for open loop simulations. As a hydrological modelling system, MESH captures many of the
important land-surface processes necessary for cold-regions simulation, provides a flexible modelling
framework that facilitates inter-comparison of alternative algorithms and models (e.g., land surface
schemes and routing schemes), and can be applied over large river basins.
Over the course of CCRN, major advancements in the MESH system were made in terms of basic
operability, scalability, and parallelization, as well as in its ability to handle sloping and complex terrain,
permafrost (Sapriza-Azuri et al., 2018; Elshamy et al., 2020), lakes and wetlands, snow processes and
glacier representation, vegetation processes including snow–canopy interactions (Bartlett and Verseghy,
2015; Asaadi et al., 2018), frozen soils and Prairie hydrology including variable hydrological connectivity
(Mekonnen et al., 2014), and water management impacts including reservoirs, diversions, and irrigation
(Yassin et al., 2019). The work has progressed to a point at which functioning MESH models for the
Mackenzie and Saskatchewan River systems have been developed, calibrated, and tested (Yassin et al.,
2017, 2019). The models have been run for historical (1980–2010) and future (2025–2055; 2070–2100)
climates at a 10 km resolution, incorporating these advancements in process and water management
representation, to examine changes in regional hydrology and river flows. Forcing data included WATCH
and ERA-Interim products with bias correction using regional datasets such as the combined Global
Environmental Multiscale (GEM) atmospheric model forecasts and the Canadian Precipitation Analysis
(CaPA) (Fortin et al., 2018). Regional climate projections for future MESH simulations to the end of the
21st century were derived from 15 ensemble members from the CORDEX-NA CanRCM4 under the RCP8.5
emissions scenario. Climate fields were spatially downscaled and bias-corrected against the WATCH ERA-
Interim reanalysis–GEM–CaPA product (Asong et al., 2020). Major efforts have been needed to develop
robust algorithms for simulation of permafrost, glacier, and vegetation change, and the development of
scenarios of future land cover change. These have now been prepared and the next phase of the work is
to run the models for full future assessment. Scenario results are currently pending, but some preliminary
insights are discussed below.

## 5. Synthesis of future change and hydrological responses

New understanding and insight into process sensitivity, interactions, and responses (Sect. 3), together
with expert elicitation and process-based modelling (Sect. 4), have allowed more scientifically-informed
projections of future ecological, cryospheric, and hydrological change than have hitherto been available.



Here, these are brought together, informed by the new research results from CCRN, to develop a summary
picture largely applicable to the late-21$^{st}$ century (Fig. 9).
Future climate is expected to lead to profound changes in land cover and vegetation.  In the mountain
regions, one of the most striking changes will be the loss of glaciers.  The lower parts of many glaciers will
have disappeared within decades or less, while upland icefields may persist, but in a much diminished
state.  By the late-21$^{st}$ century only vestigial remnants of the former ice cover and small glaciers in
favorable locations for ice preservation will likely remain.  Over a much larger part of the CCRN domain,
and of greater magnitude of change, will be the response of vegetation and forest ecosystems to climate
change and climate-induced disturbances.  At northern and alpine tundra and tree line ecotones, shrub
growth and expansion in tundra will continue and is expected to accelerate over the latter half of the 21$^{st}$
century.  A northern and upward shift in tree line is likely but will occur more slowly and be far less
pronounced than for shrub expansion.  Across the contiguous Boreal Forest, the major transition will be
the loss of ENF and major expansion of DBF and jack pine forest stands, wetlands (in the north), and to a
lesser extent, grasslands (e.g., in valley bottom areas of the Cordillera).  Permafrost thaw and collapse of
permafrost-underlain spruce forest and peat plateaus will accelerate over vast parts of the Taiga Plain.  At
the southern Boreal–Prairie ecotone and over the Boreal Plain, northward expansion of deciduous shrubs
and concomitant loss of deciduous and mixed-wood forest will continue, leading to the expansion of
grassland in these areas into the late-21$^{st}$ century.
In addition, human activities, land–water management practices, and changes in agricultural cropping
patterns will further alter landscapes.  These are likely to be most pronounced in the Prairie and southern
Boreal parts of the CCRN region.  Climate warming will further drive changes in crop mix and spatial
patterns, with new crops such as corn becoming more widespread, and northward expansion of other
crops such as canola, wheat, and soy (Hannah et al., 2020).  Climatic and land suitability limitations will
restrict how, where, and the timescales over which this occurs.  For example, parts of southern Alberta
will experience more extreme heat and heat stress days above 30°C, resulting in declining crop production
even with sufficient moisture.  In Saskatchewan, work by Coles et al. (2017) has suggested for planted
hillslopes, measured decreased snowfall, snowmelt runoff, and spring soil water content is affecting
agricultural productivity through increased dependence on growing season precipitation, likely
accentuating the future impact of droughts.  Areas vulnerable to drought, such as the Palliser Triangle of
southern Alberta and Saskatchewan, and where soils have low moisture storage capacity, will most likely
undergo conversion to pasture and grassland as arable agriculture becomes non-viable.  Other areas may
require irrigation to remain viable, and with agricultural expansion and more water-intensive forms of
crop production, there will be increased irrigation demand (Council of Canadian Academies, 2013) and
possibly a need for more reservoirs.  The northward expansion of agriculture will occur in nodes as
infrastructure and roads develop, and be limited by the suitability of soils.  Another major change in parts
of the agricultural zone is the artificial drainage of wetlands, which has various impacts on runoff, erosion,
sediment transport, groundwater recharge, and water quality (Pomeroy et al., 2014; Shook et al., 2015).
While recent polices have been implemented to limit drainage (or minimize the impacts), the trend will
likely continue, especially in wetter regions to the east and in the face of hydro-climatic change resulting
in more spring and summer flooding (Stewart et al., 2019), although the potential exists for wetland
restoration to mitigate these effects.
The combined changes in climate, vegetation, soils, and land cover will have major effects on hydrology.
CRHM outputs show that the loss of cold in the CCRN region is expected to cause dramatic shifts in the
timing, variability, and volume of streamflow, and even more profoundly, on the processes generating
streamflow.  There is sometimes compensation by changing vegetation, but also instances where


vegetation and soil change enhance the magnitude of climate change impacts on hydrology. Summary
results from the CRHM applications at several observatory basins in different ecological regions are
provided in Table 1. Results for a number of other basins are pending. These studies show a tendency
for increasing total discharge and earlier spring freshet in these headwater basins, as a result of warmer
and wetter late-21st century conditions, but mixed trends in SWE and peak discharge rates. Within
Marmot Creek, anticipated warming will cause basin-wide peak SWE to decline by about 30 to 40%, but
by as much 90% in some parts of the basin, with valley bottoms becoming almost entirely snow-free, and
an accompanying shift in snow cover depletion of up to six weeks. Yet the increase in P leads to a roughly
20% increase in total discharge. Farther north at Wolf Creek, where conditions are colder, climate change
impacts on snow regime are projected to be less severe and vegetation change (expansion of forest and
shrub tundra) is projected to have a compensatory influence. Here, a statistically insignificant increase in
SWE due to vegetation increase in the alpine zone was found to offset the statistically significant decrease
in SWE due to climate change. At high elevations in Wolf and Marmot Creeks, CHRM results indicate that
vegetation/soil changes moderate the impact of climate change on peak SWE, the timing of peak SWE,
evapotranspiration, and annual runoff volume. However, at medium elevations, these changes intensify
the impact of climate change, further decreasing peak SWE and sublimation. At Havikpak Creek near the
Taiga–Tundra transition, where significant expansion of shrubs is expected, maximum SWE will increase
as a result of increasing P and reduced blowing snow redistribution and sublimation. This is expected to
double the volume of discharge, and significantly increase spring freshet volume, snowmelt rates and
peak discharge rates.
CRHM was also applied to the Bow (~7824 km$^2$) and Elbow (~1192 km$^2$) River Basins above the city of
Calgary, AB, and run to diagnose the hydrological effects of forest disturbance in these basins in the
context of the June 2013 flood event. The land cover scenarios are at a finer resolution than those shown
in Figs. 6 and 7, but capture the same essential features and in agreement for wildfire and the loss ENF
projected for the late-21st century. Other scenarios included harvesting of lodgepole pine and disturbance
by mountain pine beetle. The results show that for both rivers, high wildfire severity and secondarily
mountain pine beetle infestation with salvage logging resulted in an increase in streamflow volume. High
wildfire severity followed by mountain pine beetle with salvage logging and maximum harvest area
scenarios increased the volume and daily discharge of the June 2013 flood. Other forest disturbance
scenarios had minimal impacts on streamflow. Thus, wildfire and loss of montane forests in such
intermediate sized basins of the mountain headwaters are likely to have a notable impact on flow regime
in future.
For the larger Saskatchewan and Mackenzie River systems, the results of MESH simulations over the
Saskatchewan and Mackenzie River Basins indicate that future climate conditions will lead to considerable
shifts in discharge timing, magnitude, and variability. The results are provisional and do not yet fully
account for changing landscapes and vegetation, but initial MESH climate production runs indicate there
is likely to be a shift in timing of spring hydrograph rise and peak flows of nearly two weeks earlier by mid-
century, and as much as one month by late-century. Fine-scale MESH runs on the mountain-sourced Bow
and Elbow River Basins, driven by WRF, and with adjustments for slope, aspect and elevation, were able
to capture the main river hydrographs well and demonstrate how this forward shift in freshet is a result
of a transition to much more rainfall-runoff generation as rainfall increases and snowpacks decline in the
late-21st century (Tessema et al., 2020). The MESH models of the Saskatchewan and Mackenzie River
Basins further show that increasing P across the CCRN region of interest is not offset by increasing ET, and
overall flow volume increases by as much as 40% by the end of the century. Low flows in winter become
slightly higher in magnitude but with more inter-annual variability, and there is a likely considerable
increase in spring freshet volume and peak flows. By late-century these spring flows, on average, will



increase by a factor of 1.5 to 2; the greater variability and higher peak flows at most locations along the
river network will greatly increase the risk of spring flooding. This is likely to stress human water
management systems and reservoir operations, as river discharge regimes may be altered far beyond the
historical flow ranges, seasonality, and variability under which these systems were designed and
operated.

## 6. Concluding remarks

This article reports results of the multi-disciplinary CCRN, which has examined recent and future
ecological, cryospheric, and hydrological change in relation to projected 21st century climatic change over
the interior of western and northern Canada. Key insights into the mechanisms and interactions of Earth
surface process responses are presented, gained from a network of highly instrumented and intensively
studied experimental observatories. This provided the ability to observe and diagnose change across the
region, while the sites acted as a testbed for developing and improving predictive models. CCRN activities
also involved improving cold region process representation within the CRHM fine-scale and MESH large-
scale modelling systems. Application of the fine-scale modelling system has been used to diagnose recent
change in selected basins, and the nature of future change. Broader application of the fine-scale and
large-scale models under future climate and land cover scenarios, representing mid- and late-21st century
conditions, is currently underway with support of the Global Water Futures program.
In general, insights from expert elicitation and preliminary modelling indicate that the region will continue
to undergo widespread environmental change as a result of warmer temperatures and changing $P$
regimes. This will predominantly involve continued loss of snow and ice, thawing of permafrost, major
ecosystem change and an increase in the occurrence and magnitude of wildfire, and a shift from nival and
glacial to more rainfall-driven pluvial runoff regimes. However, some of the process responses are non-
trivial and highly complex. To understand the trajectories of different northern ecological, cryospheric,
and hydrological systems under climate change, the details of these processes and their interactions are
very important. This can have unanticipated and sometime surprising outcomes that simple models or
extrapolations will fail to capture. Many current generation land surface schemes and hydrological
models do not handle a dynamic landscape where vegetation, glaciers, permafrost distribution, etc. are
transient, and there is large uncertainty in their application under a non-stationary hydro-climatic regime.
Human interventions also have a large influence through activities such as forest disturbance, agricultural
and forest land management, water abstractions for consumptive use, diversions, and reservoir
operations, which further alter ecological and hydrological systems.
Another critical issue relates, in part, to long-term data acquisition and organization. Climate monitoring
and observation are key to understanding its variability and trends, and for providing input to land surface
and hydrological models, yet this is a major challenge in cold regions. Forcing data remains the largest
source of uncertainty for historical simulations. In Canada, and especially in its alpine and northern
regions, there is a sparse observational network, with problems related to station automation and major
challenges associated with the measurement of solid $P$ (Rasmussen et al., 2012), thus requiring high
priority to expanding the network and to better measuring snowfall (Bush and Lemmen, 2019).
Finally, we note that modelling at multiple scales is advantageous for more fully examining Earth system
behaviour and responses. While all models have limitations, detailed process-based models can yield
important insights into interactions and feedbacks, and large-scale models can be used with careful
selection of possible scenarios to quantify likely effects of future change. The CRHM and MESH modelling



platforms provide a unique capability to represent the complex, energy-dominated processes that control
cold regions hydrology. However, while further work is underway on scenario analysis, there are also
continuing needs for the development of flexible and robust models with the capability to capture cold
region processes and bridge scales from local to regional to large basin-scale.
## Appendix: Developing Future Land-Cover Maps for Hydrologic Modelling
This Appendix describes our approach to generate future land-cover scenarios for hydrologic modeling,
based on observational and modelling studies, and expert elicitation. The scenarios were developed for
use in the MESH hydrologic model, to address the question: What is the potential for vegetation changes
to affect 21st century streamflow in the Saskatchewan and Mackenzie River basins? The approach
generated future scenarios by applying a realistic change signal to the current MESH land-cover map.
The change signal was derived from a Random Forest classification tree (RFCT) (Rehfeldt et al., 2012),
using an updated analysis from 2017. The RFCT products included a base land-cover map that was used
to represent 2005, and projected maps for 2025, 2055 and 2085 based on climate scenarios from RCP8.5.
Before computing the change signal, the RFCT vegetation classes were aggregated into nine land-cover
types that could be easily related to the MESH plant functional types (PFTs); the RFCT grid was mapped
onto the MESH grid (0.125 x 0.125 degrees); the land-cover fractions were computed for each MESH grid
square; and the 2025 and 2055 maps were averaged to represent 2040. The vegetation change signal was
then computed for each land-cover type as the difference in the fractional cover between the projected
and base maps (2040 minus 2005 and 2085 minus 2005).
The RFCT analysis did not include four of the MESH PFTs (Wetlands, Water, Ice, or Urban). Consequently,
it was necessary to limit the changes in fractional coverage to seven CLASS PFTs (Deciduous Broadleaf
Forest (DBF), Evergreen Needleleaf Forest (ENF), Mixedwood Forest (MWF, SK Basin only), Cropland,
Grassland, Shrubland, Tundra, and Barren). The Shrubland and Tundra PFTs were identical to Grassland
except for height and leaf area index. In addition, the RFCT represented prairie Grassland and Cropland
as one vegetation class, so that it was not possible to represent changes due to competition between the
two.
The resulting unmodified RFCT change signals for 2005 to 2040 and 2005 to 2085 represent the land-cover
changes that would be expected if climate was the only factor limiting vegetation migration. In reality,
vegetation migration is also limited by the rates of colonization, and in some cases, by additional
constraints such as the need for wildfire as a trigger. We used expert knowledge to eliminate unrealistic
changes from the RFCT change signal, retaining only changes that were deemed to be plausible over the
21st century. The plausible changes are listed in Table A1, with associated conditions and constraints. For
land-cover changes that normally occur only after wildfire (ENF to Grassland and ENF to DBF, Table A1),
the analysis added two further constraints. The area burned was estimated assuming a prescribed fire-
return interval which varied with latitude (Table A2). The resulting, constrained change signal represented
the maximum plausible change for each land cover type.
Finally, 2040 and 2085 projections of the MESH land cover map were created by applying the change
signal to the current MESH land-cover base maps. The main changes included:
•   to the south and west, a northward and upward (elevational) shift in the forest-grassland ecotone
45        in response to:


- o land clearing for agriculture (Cropland expansion into DBF, using the presence of DBF to indicate soils that were suitable for agriculture);
  - o partial replacement of ENF by Grassland and Shrubland following wildfire;
- within the contiguous forest, wildfire-induced partial replacement of ENF by DBF;
- at the northern and alpine tree line, displacement of Shrubland by ENF in areas where ENF is already present;
- above the northern and alpine tree lines, Shrubland expansion into Tundra.

The strategy of applying a RFCT change signal to the current land-cover map, with modifications based on constraints from expert knowledge, has several advantages over using the RFCT projections directly. It anchors the projections to the current land-cover map, potentially increasing their realism. It eliminates changes that are implausible over the modelling time frame (21$^{st}$ century). It integrates wildfire as a trigger for changes that most often occur after fire. And it preserves the characteristic patchiness of the boreal forest mosaic. Note that the resulting land-cover projections are intended for use in hydrologic modelling only; at best, they represent an informed guess of the likely changes. Caution is advised against using them in other applications.

## Data availability

Data are available through the cited sources throughout the text.

## Author contributions

Chris DeBeer led the organization and writing of the article with significant input from all co-authors on aspects of modelling, analysis, review, figures, interpretation and writing.

## Competing interests

The authors declare that they have no conflict of interest.

## Acknowledgements

We gratefully acknowledge financial support from the Natural Sciences and Engineering Research Council of Canada (NSERC) through their Climate Change and Atmospheric Research (CCAR) program. Garry Clarke provided future glacier change animations, which appear in Fig. 5. Mohamed Abdelhamed provided assistance with the vegetation scenarios in Figs. 6 and 7.

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





# Tables

**Table 1.** Summary of basin average CRHM projections of snow and discharge regime characteristics for the mid or late-21st century at various observatory basins within the CCRN domain. (NA indicates results are not available.)

| Ecological Region | WECC observatory / research basin | Maximum SWE | Snow cover duration | Snow sublimation | Spring freshet onset [centroid of flow] | Peak discharge | Total discharge | Comments | References |
|---|---|---|---|---|---|---|---|---|---|
| Western Cordillera | Marmot Creek* | –40 mm (–32%) | –49 days | –40 mm | –45 days [–12 days] | –0.12 $m^3s^{-1}$ (–11%) | +76 mm (+18%) | Study evaluated snow and hydrological responses to late-21st century climate, but did not evaluate effects of projected forest and land cover change. | Fang and Pomeroy (2020) |
| Western Cordillera | Marmot Creek* | –77 mm (–42%) | –37 days | –7 mm | –5 days [NA] | –0.02 $m^3s^{-1}$ (–3%) | +90 mm (+22%) | Study evaluated snow and hydrological responses to mid-21st century climate, as well as future vegetation and soil. Results here are responses to combined change. | Rasouli and Pomeroy (2019) |
| Boreal Cordillera | Wolf Creek | –26 mm (–20%) | –9 days | +5 mm | –22 days [NA] | –0.85 $m^3s^{-1}$ (–31%) | +36 mm (+15%) | Study evaluated snow and hydrological responses to mid-21st century climate, as well as future vegetation and soil. Results here are responses to combined change. | Rasouli and Pomeroy (2019) |
| Boreal Plain | BERMS / Whitegull Creek | –38 mm (–48%) | –59 days | NA | –25 days [–11 days] | +10 mm/day (+100%) | +37 mm (+30%) | Study comprised a sensitivity analysis to changes in P (–30% to +30%) and T (0 to +6°C), as well as forest harvesting scenarios. Results here indicate responses to +20% P and +6°C, as most closely projected by WRF for this region by late-21st century. | provisional results |
| Taiga Plain – Southern Arctic Transition | Havikpak Creek | +80 mm (+70%) | –26 days | –5 mm | –7 days [–7 days] | +0.7 $m^3s^{-1}$ (+78%) | +101 mm (+100%) | Study evaluated snow and hydrological responses to late-21st century climate, as well as future vegetation. Results here are responses to combined change. | Krogh and Pomeroy (2019) |

*Note: Difference in relative magnitude of changes for Marmot Creek are a result of differences in model base scenarios as well as projection results between the two studies.





**Table A1**. Projecting future changes in the MESH land-cover map over the 21st century: changes in the MESH plant functional types (PFT); changes in the RFCT land-covers that were used to identify areas of change; and the associated conditions and constraints. The changes were implemented separately for each MESH grid square, when all three necessary conditions (1-3) and the associated constraints were met.

| Description | Necessary Conditions | | | Projected CLASS PFT | Constraints | %Area Conversion 2005-2085 SK Basin | %Area Conversion 2005-2085 Mackenzie Basin |
| --- | --- | --- | --- | --- | --- | --- | --- |
| | 1. RFCT Land cover (base map) | 2. Projected RFCT Land cover (2040 or 2085) | 3. CLASS PFT (2005 base map) | | | | |
| Agricultural expansion into Aspen Parkland and southern boreal MWF/DBF | Aspen Parkland or Boreal MWF | Great Plains Grassland | DBF | Cropland | 80% conversion; 20% retained as DBF | 0.2% | 1.5% |
| Encroachment of Aspen Parkland into southern boreal MWF/DBF | Boreal MWF | Aspen Parkland | DBF or MWF | 50% Cropland 50% DBF | 50% conversion; 50% retained as DBF | 1.6% | 0.4% |
| Encroachment of Aspen Parkland into southern boreal ENF | Boreal MWF | Aspen Parkland | ENF | Grassland | 50% conversion; 50% retained as ENF | 0.2% | 0.2% |
| Post-fire replacement of ENF by Grassland near forest-grassland ecotone | Aspen Parkland or Boreal MWF | Great Plains Grassland | ENF | Grassland | Limited to burned area; varying conversion rate from 75% in the south (53°N) to 25% in the north (63°N) | 0.2% | 0.1% |
| Post-fire replacement of ENF by DBF in boreal ENF | Boreal MWF (no change) / Boreal ENF (no change) | | ENF | DBF | | 1.1% | 2.8% |
| Encroachment of ENF into Shrubland at tree line | Mixed ENF& Shrubland | ENF | Grassland or Shrub | ENF | Some ENF already present | NA | 0.5% |
| Shrubland expansion into Tundra | Tundra or Barren | Boreal MWF or ENF or mixed ENF/Shrubland or Shrubland | Tundra | Shrubland | None | NA | 5.0% |
| Tundra expansion into Barren | Barren | ENF or mixed ENF/Shrubland or Shrubland or Tundra | Barren | Tundra | None | NA | 2.5% |





**Table A2.** CCRN expert-guided north–south gradients in the post-fire conversion of ENF to DBF in the contiguous Boreal and Taiga Forest.

| Location | Fire Return Interval (years) | ENF Fraction Burned in 45 years | Conversion Rate | Fraction Converted from ENF To DBF |
|---|---|---|---|---|
| North (63 °N) | 120 | 31% | 25% | 8% |
| Mid (58 °N) | 100 | 36% | 50% | 18% |
| South (53 °N) | 80 | 43% | 75% | 32% |



## Figures

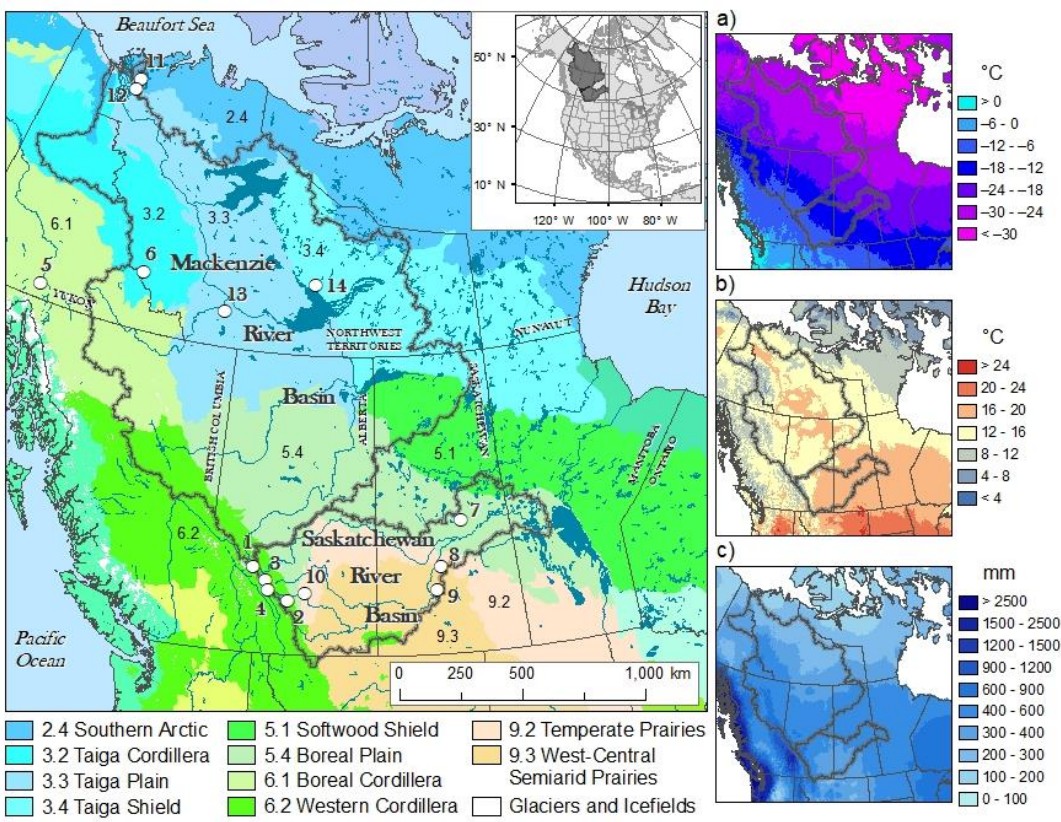

**Figure 1**. Map of the CCRN study domain across the interior of western Canada. The Mackenzie and Saskatchewan River Basins are shown and their location within North America is indicated in the inset map. Land cover and physiography are depicted by the Level II Ecological Regions of North America, with the naming convention and symbology of CEC (1997). The panels on the right show a) January mean air temperature, b) July mean air temperature, and c) annual total precipitation. The locations of CCRN Water, Ecosystem, Cryosphere, and Climate (WECC) observatories are indicated by circles: 1) Columbia Icefield, 2) Marmot Creek, 3) Peyto Glacier, 4) Lake O'Hara, 5) Wolf Creek, 5) Brintnell-Bologna Icefield, 7) Boreal Ecosystem Research and Monitoring Sites (BERMS), 8) St. Denis National Wildlife Area, 9) Brightwater Creek/Kenaston Mesonet Site, 10) West Nose Creek, 11) Trail Valley Creek, 12) Havikpak Creek, 13) Scotty Creek, 14) Baker Creek. Source data are from the North American Environmental Atlas (http://www.cec.org/sites/default/atlas/map/), the National Hydro Network (http://www.geobase.ca), WorldClim Global Climate Data (http://worldclim.org/version2), and the Commission for Environmental Cooperation (http://www.cec.org).



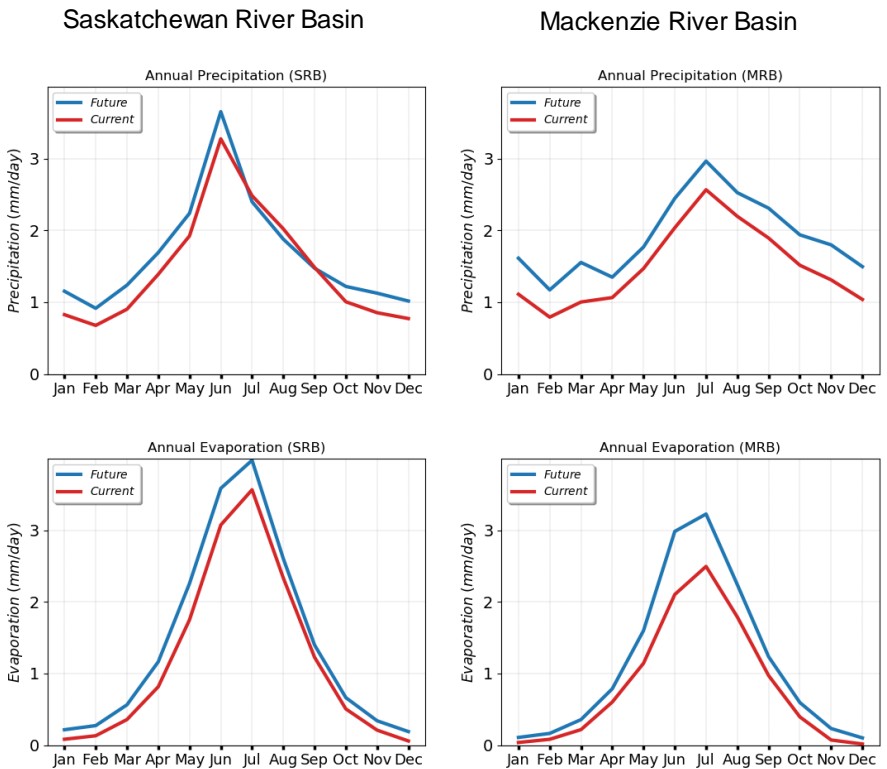

**Figure 2**. Simulated *P* and ET surface water budget components (mm day$^{-1}$) over the Saskatchewan (left) and Mackenzie (right) River Basins for the WRF control (current; 2000–2015) and future (2085–2100) periods. Results are from Kurkute et al. (2020).





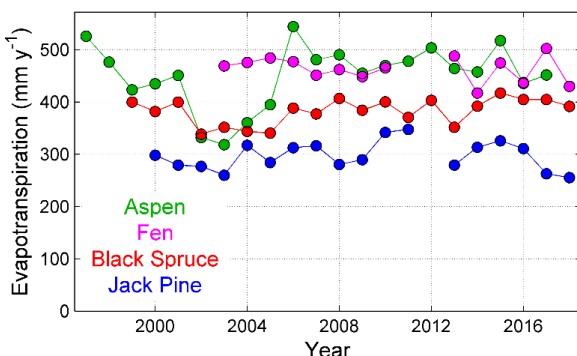

**Figure 3.** Annual ET (with energy-balance-closure adjustments) at the four BERMS sites from 1997 to 2018 showing generally higher values for the Old Aspen site than for the two conifer sites. The dry conifer site (Old Jack Pine) generally had lower ET than the wet conifer site (Old Black Spruce). The Fen had values exceeding the Old Aspen site following the 2001-2003 drought, with similar values in other years.





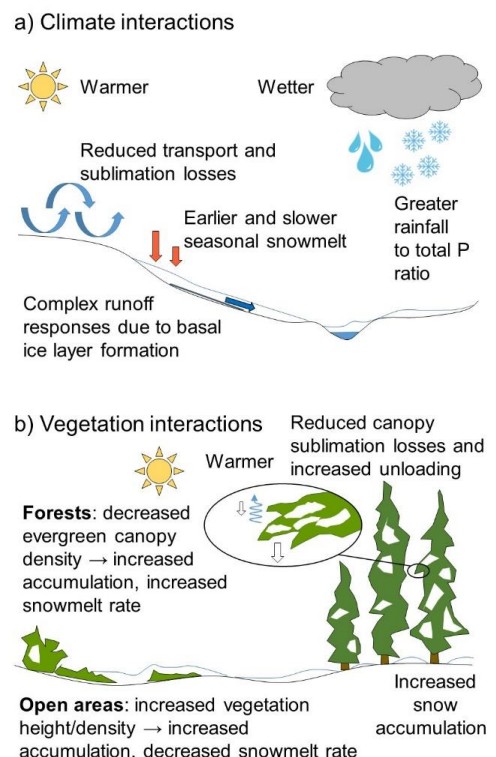

**Figure 4.** Conceptual schematic of expected snow change in the CCRN domain and similar cold regions. Warmer conditions lead to less snow while wetter conditions can lead to more or less snow; warmer and wetter conditions can be partially compensatory. Other changes complicate the snow–climate interactions, and spatial patterns of vegetation change with respect to snow processes control snow response.



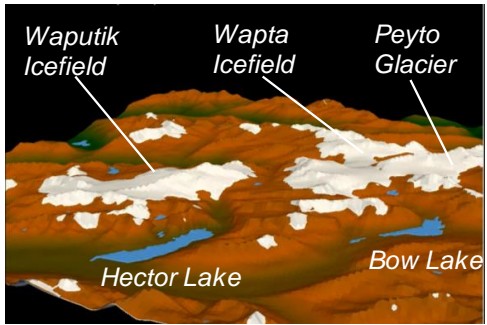

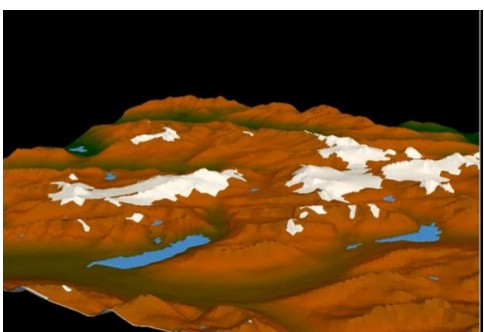

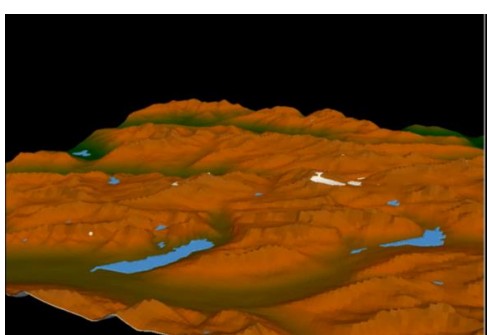

**Figure 5.** Simulated glacier projections in 3D perspective along the continental divide and in the headwaters of the Saskatchewan River for a) 2005, b) 2040, and c) 2085 using the CanESM RCM under the RCP8.5 forcing scenario. Scale varies in the perspective, but the ground distance across the length of the Waputik Icefield in the 2005 scene is roughly 12 km. Results are from Clarke et al. (2015).





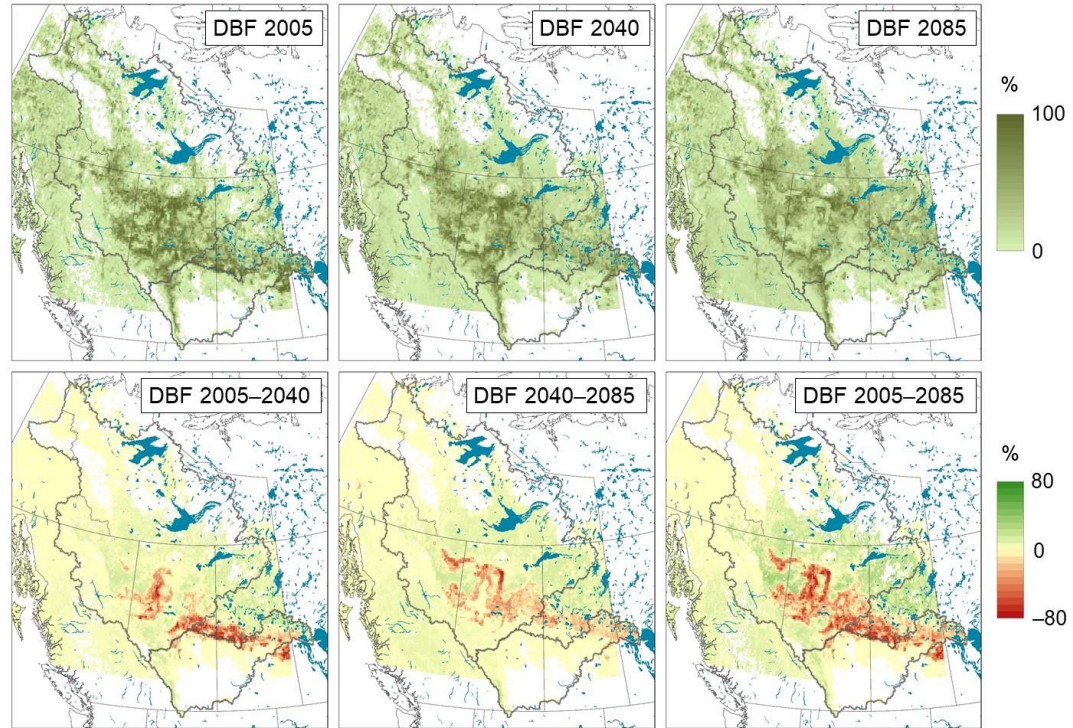

**Figure 6.** Changing DBF cover fractions over the Mackenzie and Saskatchewan River Basins in the 21st century. The approach involved a simple, yet ecologically-based projection with expert-guided modifications to impose restrictions on the rates of species colonization and requirements for wildfire to trigger change (Appendix). Projections were made in 45-year increments from the base period (centered at 1995, but using the 2005 base map) to represent the 2040 (mid-century) and 2085 (late-century) periods.

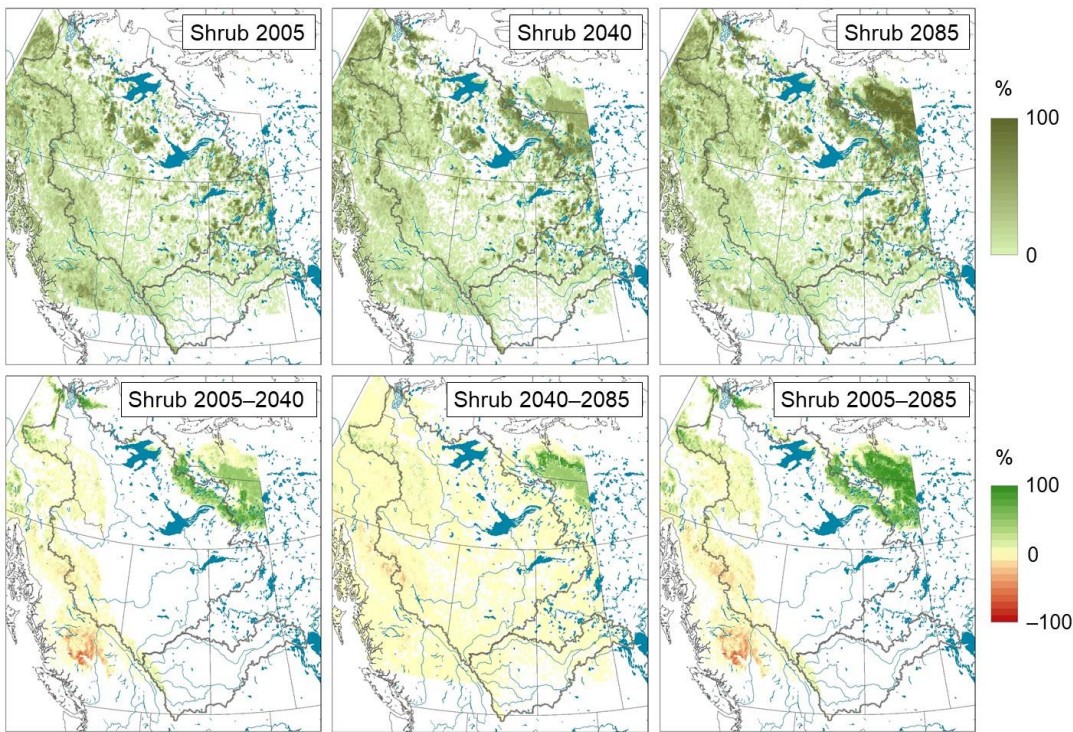

**Figure 7.** Changing shrub cover fractions over the Mackenzie and Saskatchewan River Basins in the 21st century derived from CCRN expert-guided modifications to climate-based projections using the methodology of Rehfeldt et al. (2012) (Appendix). Projections were made in 45-year increments from the base period (centered at 1995, but using the 2005 base map) to represent the 2040 (mid-century) and 2085 (late-century) periods.



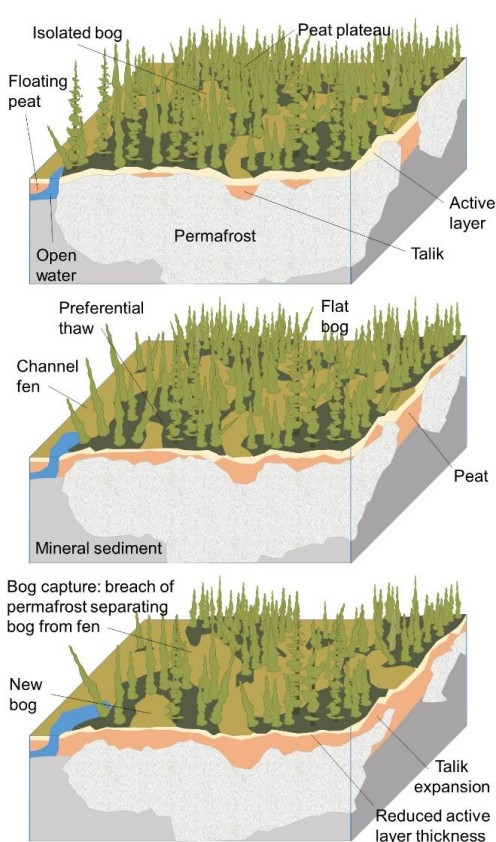

**Figure 8.** Conceptual model of forest canopy thinning and permafrost thaw in the Taiga Plain, after Quinton et al. (2009; 2019) and Connon et al. (2018).



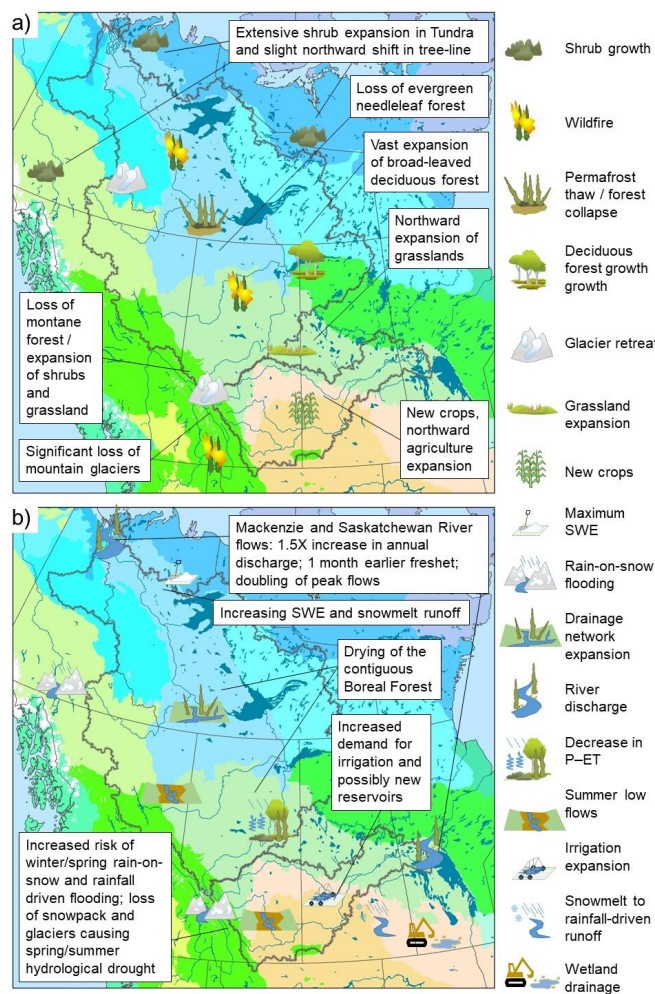

**Figure 9.** Conceptual depiction and synthesis of surface changes over the CCRN region, by the late-21st century, for a) land-cover and vegetation, b) hydrological regime and water management. The base map depicts the Level II Ecological Regions of North America as shown in Fig. 1.