# Peer review of "Summary and synthesis of Changing Cold Regions Network (CCRN)"

_Hydrology and Earth System Sciences, 2020_

## Referee Comment (RC1) · Anonymous Referee #1 · 23 Nov 2020

General comments This paper reports results of the multi-disciplinary CCRN, which has examined recent and future ecological, cryospheric, and hydrological change in relation to projected 21st century climatic change over the interior of western and northern Canada. Based on field studies and understanding from the observatories, key insights into the mechanisms and interactions of Earth surface process responses are presented, as well as the developed fine-scale and large-scale hydrological models and their projected results. I think this is a monumental summary work in the western and northern Canada, and it is also benefit for the development of Earth surface process

research in the world. In addition, this manuscript is well-written and technical sound, and the topic is interesting for the readers of Hydrology and Earth System Sciences. I recommend publication after minor revision.

Specific comments Page 9, Line 46: Just an advice: Here only the results under RCP 8.5 are described, other results may need to be presented such as under RCP 4.5, or 1.5 or 2.0 degree temperature rise scenarios that would happen more possibly. Page 13, Line 4-5: As a result of warming and shallower re-freeze depths during winter, active layer thickness has been decreasing. Please check.

Technical corrections Page 8, Line 30, "a nd" should be "and"

---

## Referee Comment (RC2) · Francesco Avanzi (Referee) · 5 Dec 2020

DeBeer et al present an exhaustive overview of future changes to the Critical Zone of NW Canada based on an interdisciplinary initiative, the Changing Cold Regions Network (CCRN). This overview comprises a rather detailed discussion of mechanisms driving impacts of a changing climate on precipitation, evapotranspiration, vegetation distribution and type, snow and ice accumulation and melt, and sub-surface water recharge, among others. The paper also describes advances made by CCRN in terms of new modeling approaches, and ultimately summarizes predicted future changes

based on the gained new understanding and the improved models.

I concur with the other Reviewer that this is an interesting, substantial, and well written manuscript. Despite having only tangential knowledge of Canadian hydrology and more in general of high-latitude processes, I was able to follow the main line of reasoning with only occasional need for clarifications (see below). Thus, I recommend the editor to accept this paper after minor revisions, as outlined below.

GENERAL COMMENTS

1. According to HESS guidelines, a review paper should "summarize the status of knowledge and outline future directions of research within the journal scope" (https://www.hydrology-and-earth-system-sciences.net/about/manuscript_types.html). The paper is excellent in the way it summarizes the status of knowledge, but I think it would be even a more substantial contribution to the current literature if it included a specific section on future directions of research. What is left to understand after the substantial – and unifying – contribution provided by CCRN? Are there significant knowledge gaps that authors would recommend for future initiatives? Some of these knowledge gaps are already mentioned throughout the manuscript, but being more explicit on this in a dedicated section would sharpen the message of the paper and further enhance its impact.

2. I was intrigued by the following statement in the abstract and somewhere else in the main text: "It is essential to consider the underlying processes and base predictive models on the proper physics, especially under conditions of non-stationarity where the past is no longer a reliable guide to the future and system trajectories can be unexpected." I can only agree with this statement, as it points to the recurring hypothesis that improving process representation in models will ultimately translate into increased realism and performance. While this may appear axiomatic, in fact issues with equifinality and conceptual uncertainty in hydrology have often challenged this hypothesis. So I was wondering if authors could enrich their section about modeling improvements

with some examples of cases when they related model failure to its failure to consider underlying processes, and possibly examples of cases when they succedeed in improving model performance by increasing its realism based on the numerous study plots and datasets they have collected. From a modeling point of view, doing so would add another compelling argument to this paper.

3. Relatedly, section 4 could be enriched by one paragraph discussing performance of these models for the current climate, given that the following Section 5 discusses future scenarios based on this modeling (is my understanding correct?).

SPECIFIC COMMENTS

- line 25 page 2: what do you mean with "freashwater ice cover"?

- line 41 page 2: see general comment 1

- line 21 page 4: maybe report some statistics here to give an idea of aridity?

- line 12-13 page 8: I missed an explanation of why in the Arctic earlier AND FASTER melt is predicted.

- Line 26ff page 8: rising temperatures may also alter the way long-wave radiation delays - or accelerates – melt (https://agupubs.onlinelibrary.wiley.com/doi/full/10.1002/wrcr.20504). Consider including this in details as you did at line 48 for shrub.

---

## Author Comment (AC1) · 17 Dec 2020

Author's Response

We thank the two reviewers for their time and effort and for the helpful comments on the manuscript. The positive feedback is very much appreciated. We will make a concerted effort to address these points and revise the manuscript. Below we respond to the individual comments in detail (in red font).

Reviewer #1

Specific comments

Page 9, Line 46: Just an advice: Here only the results under RCP 8.5 are described, other results may need to be presented such as under RCP 4.5, or 1.5 or 2.0 degree temperature rise scenarios that would happen more possibly.

Clarke et al. (2015) do indeed present glacier change results for other RCPs and we will make sure to include a summary of their projections under even the more modest future climate warming scenarios.

Page 13, Line 4-5: As a result of warming and shallower re-freeze depths during winter, active layer thickness has been decreasing. Please check.

Thank you for raising this point. This involves 1) permafrost terminology, and 2) active layer processes. Active layer is the zone that thaws AND freezes each year. Warming leads to increased thickness of the suprapermafrost layer (suprapermafrost layer includes all layers above the permafrost table: i.e. the active layer (seasonally frozen) and the talik (perennially thawed).

Technical corrections Page 8, Line 30, "a nd" should be "and"

We will correct this.

Reviewer #2

GENERAL COMMENTS

1. According to HESS guidelines, a review paper should "summarize the status of knowledge and outline future directions of research within the journal scope" (https://www.hydrology-and-earth-system-sciences.net/about/manuscript_types.html). The paper is excellent in the way it summarizes the status of knowledge, but I think it would be even a more substantial contribution to the current literature if it included a specific section on future directions of research. What is left to understand after the substantial – and unifying – contribution provided by CCRN? Are there significant knowledge gaps that authors would recommend for future initiatives? Some of these knowledge gaps are already mentioned throughout the manuscript, but being more explicit on this in a dedicated section would sharpen the message of the paper and further enhance its impact.

This is a useful suggestion and we will follow the advice in our revision. We will include an explicit discussion on the knowledge gaps and priority issues for future research.

2. I was intrigued by the following statement in the abstract and somewhere else in the main text: "It is essential to consider the underlying processes and base predictive models on the proper physics, especially under conditions of non-stationarity where the past is no longer a reliable guide to the future and system trajectories can be unexpected." I can only agree with this statement, as it points to the recurring hypothesis that improving process representation in models will ultimately translate into increased realism and performance. While this may appear axiomatic, in fact issues with equifinality and conceptual uncertainty in hydrology have often challenged this hypothesis. So I was wondering if authors could enrich their section about modeling improvements with some examples of cases when they related model failure to its failure to consider underlying processes, and possibly examples of cases when they succedeed in improving model performance by increasing its realism based on the numerous study plots and datasets they have collected. From a modeling point of view, doing so would add another compelling argument to this paper.

We agree and feel this would strengthen the arguments in the paper about improving model performance through improving process representation. There are numerous clear examples in the published literature to draw from, particularly using the CRHM and MESH models. In our revision, we will include instances where new or improved process representations have led to more successful model simulations, using the CCRN work on various processes as examples.

3. Relatedly, section 4 could be enriched by one paragraph discussing performance of these models for the current climate, given that the following Section 5 discusses future scenarios based on this modeling (is my understanding correct?).

We will include a brief qualitative assessment of the model performance for historical simulations, but will reserve more detailed treatment of this for one or more full papers in due course.

SPECIFIC COMMENTS

- line 25 page 2: what do you mean with "freashwater ice cover"?

Here we mean ice cover on lakes and rivers. We will revise to make this clear.

- line 41 page 2: see general comment 1

We will address general comment #1 as noted above. Thank you for the suggestion.

- line 21 page 4: maybe report some statistics here to give an idea of aridity?

Yes, we will do this.

- line 12-13 page 8: I missed an explanation of why in the Arctic earlier AND FASTER melt is predicted.

We are making the point that the CRHM results differ at the Arctic site from those at more southerly locations. There is also an elevation dependency in the results and we will be more clear about this in the revised manuscript.

- Line 26ff page 8: rising temperatures may also alter the way long-wave radiation delays - or accelerates – melt (https://agupubs.onlinelibrary.wiley.com/doi/full/10.1002/wrcr.20504). Consider including this in details as you did at line 48 for shrub.

Thank you for this reference. We will review this in detail and determine if it is relevant to our paper. We will certainly include longwave radiation as a component of the energy balance and the impact of its changes under a warming climate as a topic of the discussion.

---

## Author Comment (AC2) · 17 Dec 2020

Thank you for the time and effort and for the helpful comments on the manuscript. Please see the supplemental file for our responses.

Please also note the supplement to this comment: https://hess.copernicus.org/preprints/hess-2020-491/hess-2020-491-AC2-supplement.pdf

---

## Author Response (AR1)

Author's Response

We thank the two reviewers for their time and effort and for the helpful comments on the manuscript. The positive feedback is very much appreciated. We have made a concerted effort to address these points and revise the manuscript. Below we respond to the individual comments in detail (in red font). We have also made a number of other minor edits to the paper, as can be found throughout indicated by track changes, and have added more references. None of these change any of the substance of the paper, but rather, they simply improve the syntax. Finally, we have added three more co-authors as they each made important contributions.

Reviewer #1
Specific comments
Page 9, Line 46: Just an advice: Here only the results under RCP 8.5 are described, other results may need to be presented such as under RCP 4.5, or 1.5 or 2.0 degree temperature rise scenarios that would happen more possibly.
Clarke et al. (2015) do indeed present glacier change results for other RCPs and we have now included a range of their projections to show even the more modest future climate warming scenarios.
Page 13, Line 4-5: As a result of warming and shallower re-freeze depths during winter, active layer thickness has been decreasing. Please check.
Thank you for raising this point. This involves 1) permafrost terminology, and 2) active layer processes. Active layer is the zone that thaws AND freezes each year. Warming leads to increased thickness of the suprapermafrost layer (suprapermafrost layer includes all layers above the permafrost table: i.e. the active layer (seasonally frozen) and the talik (perennially thawed).
Technical corrections Page 8, Line 30, "a nd" should be "and"
We have corrected this.

Reviewer #2
GENERAL COMMENTS
1. According to HESS guidelines, a review paper should "summarize the status of knowledge and outline future directions of research within the journal scope" (https://www.hydrology-and-earth-system-sciences.net/about/manuscript_types.html). The paper is excellent in the way it summarizes the status of knowledge, but I think it would be even a more substantial contribution to the current literature if it included a specific section on future directions of research. What is left to understand after the substantial – and unifying – contribution provided by CCRN? Are there significant knowledge gaps that authors would recommend for future initiatives? Some of these knowledge gaps are already mentioned throughout the manuscript, but being more explicit on this in a dedicated section would sharpen the message of the paper and further enhance its impact.
This is a useful suggestion and we have followed the advice in our revision. We include an explicit discussion on the knowledge gaps and priority issues for future research at the end of the paper. We have renamed that section "Further research priorities and concluding remarks".
2. I was intrigued by the following statement in the abstract and somewhere else in the main text: "It is essential to consider the underlying processes and base predictive models on the proper physics, especially under conditions of non-stationarity where the past is no longer a reliable guide to the future and system trajectories can be unexpected." I can only agree with this statement, as it points to the recurring hypothesis that improving process representation in models will ultimately translate into increased realism and performance. While this may appear axiomatic, in fact issues with equifinality and conceptual uncertainty in hydrology have often challenged this hypothesis. So I was wondering if authors could enrich their section about modeling improvements with some examples of cases when they related model failure to its failure to consider underlying processes, and possibly examples of cases

when they succedeed in improving model performance by increasing its realism based on the numerous study plots and datasets they have collected. From a modeling point of view, doing so would add another compelling argument to this paper.

We agree and feel this would strengthen the arguments in the paper about improving model performance through improving process representation.  In our revision we have included a short discussion in Section 4.1 of a key study using the CRHM platform at Marmot Creek.  This showed how including new and improved process representations led to more successful model simulations.

3. Relatedly, section 4 could be enriched by one paragraph discussing performance of these models for the current climate, given that the following Section 5 discusses future scenarios based on this modeling (is my understanding correct?).

We now include a brief qualitative assessment of the model performance for historical simulations, but will reserve more detailed treatment of this for one or more full papers in due course.

SPECIFIC COMMENTS

- line 25 page 2: what do you mean with "freashwater ice cover"?

Here we mean ice cover on lakes and rivers.  We have revised to make this clear.

- line 41 page 2: see general comment 1

We have addressed general comment #1 as noted above.  Thank you for the suggestion.

- line 21 page 4: maybe report some statistics here to give an idea of aridity?

We have included total annual P to give a sense of aridity.

- line 12-13 page 8: I missed an explanation of why in the Arctic earlier AND FASTER melt is predicted.

We are making the point that the CRHM results differ at the Arctic site from those at more southerly locations.  We have tried to be more clear about this in the revised manuscript.

- Line 26ff page 8: rising temperatures may also alter the way long-wave radiation delays - or accelerates – melt (https://agupubs.onlinelibrary.wiley.com/doi/full/10.1002/wrcr.20504). Consider including this in details as you did at line 48 for shrub.

Thank you for this reference.  We have included it and added a comment on forests and the effects of changing influence of longwave radiation under climate warming.